# High-throughput fabrication of soft magneto-origami machines

Shengzhu Yi[1,7], Liu Wang [2,7], Zhipeng Chen[1,7], Jian Wang[1], Xingyi Song[1], Pengfei Liu[1], Yuanxi Zhang[1], Qingqing Luo[1], Lelun Peng[1], Zhigang Wu[3✉], Chuan Fei Guo [4,5,6✉] & Lelun Jiang [1✉]

Soft magneto-active machines capable of magnetically controllable shape-morphing and locomotion have diverse promising applications such as untethered biomedical robots. However, existing soft magneto-active machines often have simple structures with limited functionalities and do not grant high-throughput production due to the convoluted fabrication technology. Here, we propose a facile fabrication strategy that transforms 2D magnetic sheets into 3D soft magneto-active machines with customized geometries by incorporating origami folding. Based on automated roll-to-roll processing, this approach allows for the high-throughput fabrication of soft magneto-origami machines with a variety of characteristics, including large-magnitude deploying, sequential folding into predesigned shapes, and multi-variant actuation modes (e.g., contraction, bending, rotation, and rolling locomotion). We leverage these abilities to demonstrate a few potential applications: an electronic robot capable of on-demand deploying and wireless charging, a mechanical 8-3 encoder, a quadruped robot for cargo-release tasks, and a magneto-origami arts/craft. Our work contributes for the high-throughput fabrication of soft magneto-active machines with multi-functionalities.

[1] Guangdong Provincial Key Laboratory of Sensor Technology and Biomedical Instrument, School of Biomedical Engineering, Sun Yat-Sen University, 518107 Shenzhen, China. [2] CAS Key Laboratory of Mechanical Behavior and Design of Materials, Department of Modern Mechanics, University of Science and Technology of China, 230026 Hefei, Anhui, China. [3] State Key Laboratory of Digital Manufacturing Equipment and Technology, Huazhong University of Science and Technology, 430074 Wuhan, China. [4] Department of Materials Science and Engineering, Southern University of Science and Technology, 518055 Shenzhen, China. [5] Shenzhen Engineering Research Center for Novel Electronic Information Materials and Devices, Southern University of Science and Technology, 518055 Shenzhen, Guangdong, China. [6] Guangdong Provincial Key Laboratory of Functional Oxide Materials and Devices, Southern University of Science and Technology, 518055 Shenzhen, Guangdong, China. [7] These authors contributed equally: Shengzhu Yi, Liu Wang, Zhipeng Chen. ✉email: zgwu@hust.edu.cn; guocf@sustech.edu.cn; jianglel@mail.sysu.edu.cn

Soft active machines capable of shape-morphing and locomotion in response to external stimuli (e.g., temperature, pH, light, magnetic and electrical fields) hold great promise in diverse fields, such as miniature surgical devices[1], actuator[2,3], soft robots[4,5], and flexible electronics[6–8]. Among others, magnetic actuation has been widely adopted due to its untethered control, fast response, and large penetration range[9–11]. Although various strategies for fabricating soft magneto-active machines have been proposed (e.g., template molding[12], extrusion-based 3D printing[13–15], laser printing[16,17], voxel assembly[18], and transfer printing[19]), many existing fabrication methods either can achieve low structural complexity, or they are a time-consuming process (Supplementary Table 1). For example, the template molding method may only make 2D structures[12,20] and the extrusion-based 3D printing fails to print complex structures due to the viscosity and the die swell of the composite ink[21]. The laser programming method usually requires a relatively long time to encode magnetic polarity patterns in a large area[16,22]. Therefore, the fast fabrication of soft magneto-active machines with customized architectures yet remains an unresolved challenge.

Due to the excellent customizability, origami has been widely employed to construct 3D structures[23,24]. By harnessing the folding process, origami transforms a planar sheet into a 3D architecture, imparting the intrinsic shape programmability and shape-morphing capability[25]. When integrating with stimuli-responsiveness, origami structures have shown great potential in developing functional and autonomous systems, which offers a window for designing soft magneto-active machines[20,26,27]. A simple way to empower origami with magnetic responsiveness is to directly adhere permanent magnets to the origami[28]. For example, Rus's group has created a few magnet-based origami robots and demonstrated the capability of locomotion[29], metamorphosis[30], swimming[31], and drug delivery[32]. However, the presence of permanent magnets limits the softness and miniaturization of the robot, while increasing the undesired risk of magnet break-off in clinic applications[33]. Recently, Song et al. developed a soft magnetic sheet that can be folded into various magneto-origami structures[34]. But without a backing layer, such a pure magnetic sheet exhibits an inferior structural resolution to its counterpart of paper origami, especially at folding creases[34]. Yang et al. sprayed uncured magnetic composite to the origami, yet still experienced prolonged time in sequentially curing/magnetizing each part of the origami[35]. So far, no fabrication method leverages structural complexity, softness, resolution, and fast production of magneto-origami machines at a large scale.

Here, we propose a facile fabrication strategy that can rapidly construct soft magneto-active machines at a large scale by incorporating roll-to-roll processing of 2D patterns and 3D origami folding (Fig. 1). By simply coating and curing a layer of magnetic composite (i.e., dispersing hard-magnetic microparticles in the polymer matrix) on a piece of raw paper, we create a paper-like soft magnetic sheet that can be folded into magneto-origami machines with customized geometries and high resolution. The magnetic polarity of the magneto-origami machine is quickly encoded at one time by a strong impulse magnetizing field (H ~3 T). When subjected to an actuation magnetic field, the magneto-origami machine can achieve on-demand shape-morphing. We demonstrate a set of magneto-origami machines that can fold, bend, roll, and walk with potential applications of deployment, object manipulation, and locomotion. Compared with existing soft magneto-active machines, our roll-to-roll platform allows for the automated fabrication of 2D magnetic patterns in combination with a quick magnetization process and folding, giving rise to facile and high-throughput production of magneto-active machines in various geometries and multimodal shape-morphing capabilities (Supplementary Table 1).

## Results

**Fabrication of soft magneto-origami machines.** The magnetic composite consists of magnetizable neodymium-iron-boron (NdFeB) particles (average diameter ~30 μm) embedded in the polymer resin (i.e., Ecoflex 00-10, Smooth-On) with a volume fraction of 25%. The 2D magnetic sheet is made by coating and curing a layer of magnetic composite on a piece of raw paper (Fig. 1a and Supplementary Fig. 1). To achieve an automated fabrication of 2D magnetic sheets, we build a roll-to-roll platform as shown in Fig. 1b–e. The major components of roll-to-roll platform consist of a roller of raw paper, a scraper (Fig. 1c) that creates an even layer of uncured magnetic composite on the raw paper, a heater (Fig. 1e) that cures the magnetic composite layer, a laser machine (Fig. 1d) that cuts the magnetic sheet into designed patterns, and an end roller. After curing, the designed 2D pattern is cut using a laser with uncut connections to allow for an easy tear-off (Supplementary Fig. 1). The rolling velocity, the temperature of the heater, and the pattern of the magnetic sheet in the roll-to-roll process can be controlled in an automated manner, which lays the foundation for the high-throughput fabrication of magneto-origami machines (see "Methods"). Supplementary Movie 1 shows the detailed roll-to-roll process in which the production rate is 100 mm min$^{-1}$.

Next, the torn 2D pattern can be folded into a 3D origami. Each fold is first 180° folded to make consistent foldings. (Fig. 1a). To endow the magnetic response, the folded origami was then magnetized to saturation by an impulse magnetic field (H ~3 T) (Fig. 1a), yielding a residual magnetization **M** with a strength of 170 kA m$^{-1}$ due to the hard-magnetic nature of the NdFeB particles (Supplementary Fig. 2)[36,37]. Thereafter, the magnetizing field and folding forces are removed and the 180° folded origami recovers partially to the rest state to form the magneto-origami machine. To maintain the desired shape of the magneto-origami machine (e.g., to avoid shape collapse or excessive recovery), the mechanical properties of the magnetic sheet are carefully designed. Different recipes of magnetic sheets are tested by coating magnetic composite on aluminum (Al) foil, polyimide (PI) membrane, and raw paper (Supplementary Fig. 3). It is found that the Al-based magnetic sheet is too floppy to sustain the shape well (similar to the backing-free magnetic sheet developed by Song et al.[34]), while the PI-based magnetic sheet is too stiff to be folded. By coating a 100-μm-thick magnetic composite layer on top of a 90-μm-thick raw paper (Fig. 1f and Supplementary Fig. 4), the paper-based magnetic sheet (Young's modulus ~7 MPa, ultimate tensile stress ~14 MPa, Supplementary Fig. 5) preserves the excellent folding capability of paper, allowing for the consistent fabrication of magneto-origami machines in a broad spectrum of geometries with a high structural resolution, as showcased in Fig. 1g. Notably, the folded magnetic sheet behaves like an elastic hinge (Supplementary Fig. 6) and shows a negligible change in the bending angle after 1000 cycles of folding and unfolding (Supplementary Fig. 7). Such an excellent folding/unfolding reversibility makes it a promising candidate for creating magneto-origami machines that can be actuated multiple times with consistent performance.

**Shape-morphing mechanism of magneto-origami machines.** Fully magnetized magneto-origami machines retain the residual magnetic polarity patterns, i.e., magnetization **M**. When subjected to an actuation magnetic field, magneto-origami machines can deform and move. In this work, we select a cuboid permanent magnet (NdFeB, dimension: 50 × 25 × 20 mm, residual magnetization 10$^3$ kA m$^{-1}$) as the magnetic actuation source (Supplementary Fig. 8) because it is easier to handle and can generate stronger magnetic fields (up to 200 mT in this work) than

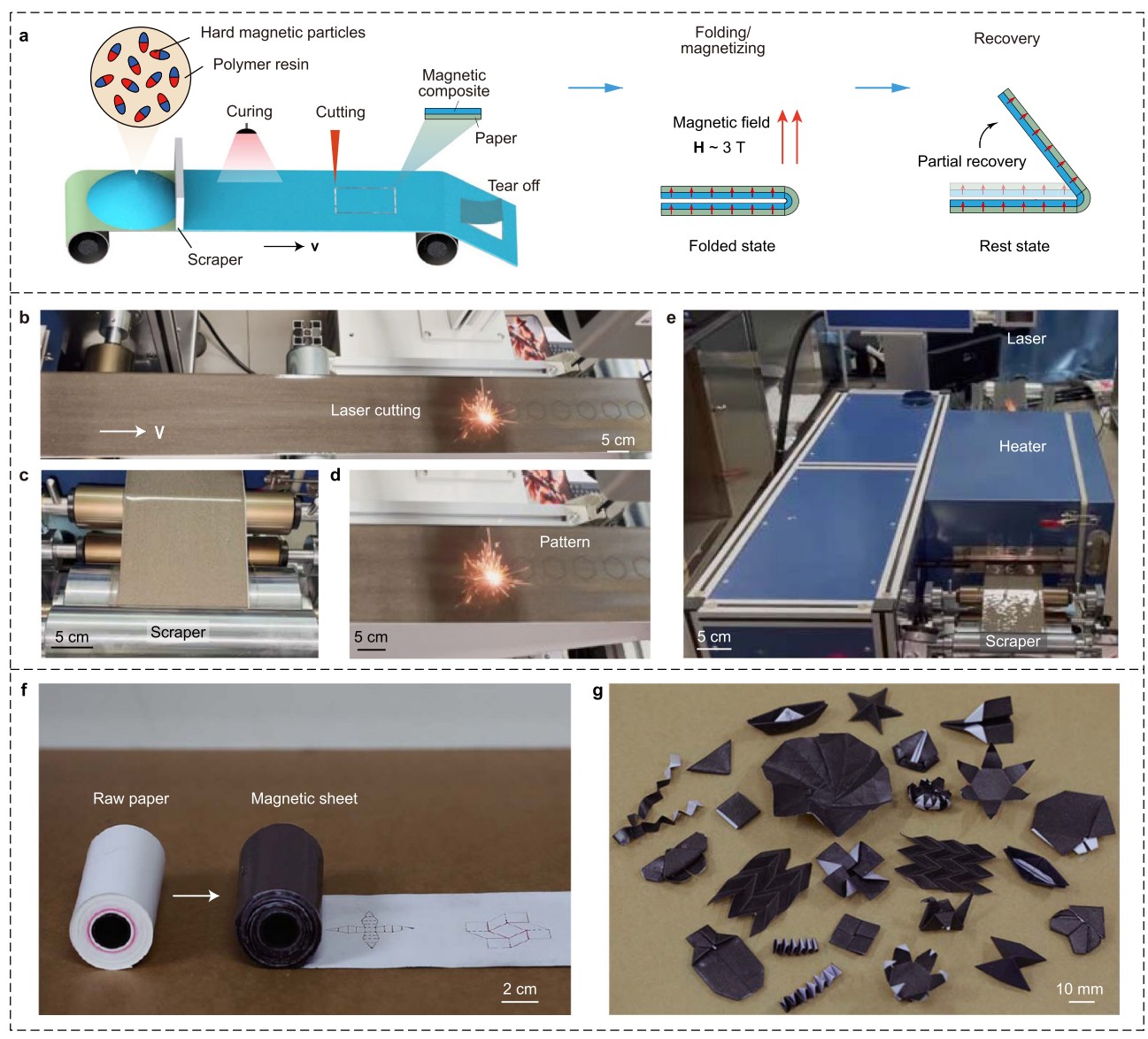

**Fig. 1 Fabrication of magneto-origami machines. a** Schematic illustration of fabrication processes of the patterned magnetic sheet by roll-to-roll method, folding/magnetizing, and partial recovery. **b–e** Experimental pictures of (**b**) laser cutting, (**c**) scraper, (**d**) pattern, and (**e**) curing heater of the roll-to-roll platform to fabricate patterned magnetic sheets. **f** Images showing a roll of raw paper and a magnetic sheet with a 2D origami pattern. **g** Magneto-origami machines with customized geometries and high structural resolution can be consistently fabricated.

electromagnetic coils. By varying the position/orientation of the cuboid magnet, the magnetic field strength **B** around the magneto-origami machine can be effectively changed (Supplementary Fig. 8). For example, the magnetic field strength on the centerline of the cuboid magnet can reach up to 200 mT when the distance between the center of the magnet and the magneto-origami machine is around 22.5 mm (Supplementary Fig. 8).

To elucidate the magnetically controllable shape-morphing mechanism, here we adopt a single-crease magneto-origami machine as a representative example (Fig. 2a). Upon actuation, the upper panel of the magneto-origami machine rotates around the crease due to the driving force, i.e., the magnetic torque density $\mathbf{M} \times \mathbf{B}$ and body force $(\nabla \mathbf{B})\mathbf{M}$ where operator "$\times$" and "$\nabla$" represents the cross-product and gradient, respectively. Therefore, the total magnetic torque that deforms the upper panel can be calculated as

$$\mathbf{T}_{\mathrm{m}} = \int_{\Omega} [\mathbf{M} \times \mathbf{B} + \mathbf{r} \times (\nabla \mathbf{B})\mathbf{M}] \, \mathrm{d}v \qquad (1)$$

where $\Omega$ represents the 3D domain of the upper panel and $\mathbf{r}$ is the position vector of a spatial point on the upper panel with respect to the crease (Fig. 2a). Based on our experimental measurement, the folding crease behaves like an elastic hinge in which the total elastic torque is linearly proportional to angle change of the crease, i.e.,

$$\mathbf{T}_{\mathrm{e}} = kL(\beta - \beta_0) \qquad (2)$$

where $k = 1.47\,\mu\mathrm{N/deg}$ represents the torsional constant of the crease per unit width, $L$, $\beta_0$, and $\beta$ denote the width, the rest angle, and the deformed angle of single-crease magneto-origami, respectively (Fig. 2a and Supplementary Fig. 6). Therefore, the deformed angle $\beta$ can be found when the equilibrium is reached, i.e.,

$$\mathbf{T}_{\mathrm{m}} = \mathbf{T}_{\mathrm{e}} \qquad (3)$$

Considering that magnetic fields around a cuboid magnet are nonuniform, Eq. (3) is solved by finite element analysis (FEA)

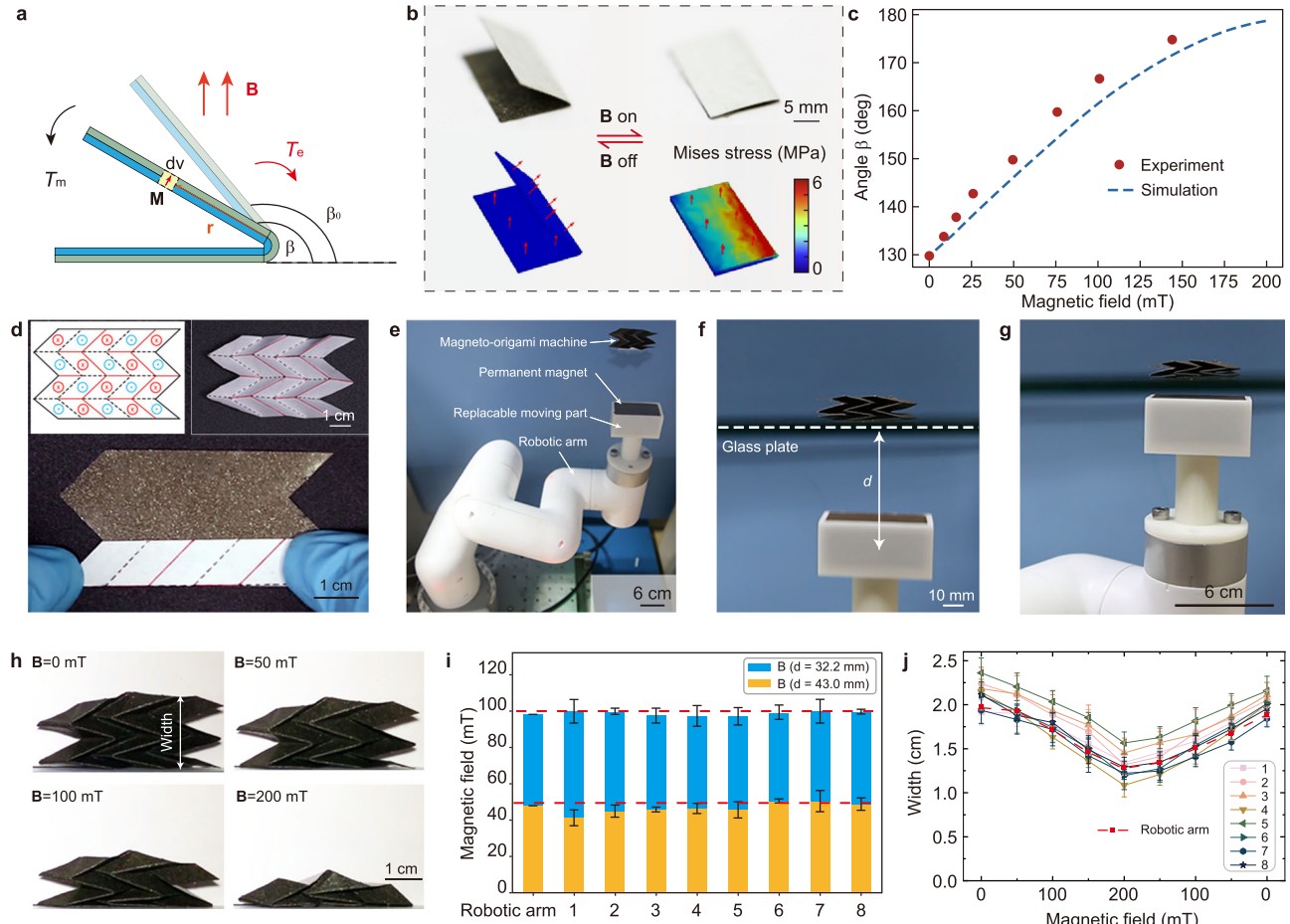

**Fig. 2 Theoretical analysis, finite element simulations, and reproductivity test of magneto-origami machines. a** Schematic illustration showing the shape-morphing of the single-crease magneto-origami actuated by a magnetic field. Equilibrium is reached when the total driving magnetic torque $T_m$ is equal to elastic torque $T_e$. **b** Comparison of deformation of single-crease origami between experiments and finite element simulations. **c** Rotation angle $\beta$ as a function of applied magnetic field strength up to 200 mT. **d** Example of folding Miura magneto-origami from the 2D magnetic sheet, in which red solid lines represent the mountain folds while black dashed lines denote valley folds. The left insert shows the magnetic polarity pattern where cross means magnetization pointing inward while blue represents magnetization pointing outward. The right insert shows the finalized Miura magneto-origami. **e** Manipulating Miura magneto-origami machine by a robotic arm. **f** By changing the actuation distance (denoted as $d$) with a robotic arm, the magnetic fields at the Miura magneto-origami machine can be accurately tuned. **g** Reducing the actuation distance $d$, the Miura magneto-origami machine shrinks. **h** Demonstration of shape-morphing of Miura magneto-origami under different magnetic fields. **i** Comparison between the measured magnetic field strength ± standard errors between the robotic arm and human hands of eight volunteers at actuation distance $d = 32.2$ and $43.0$ mm, respectively. **j** Statistical analysis of reproductivity and performance evaluation of Miura magneto-origami machines fabricated by eight volunteers each of whom folded seven samples. Each solid curve represents the mean width of seven samples fabricated by a specific volunteer who manually operated the magnet. The error bar shows the width variation of the seven samples. The dashed black curve shows the mean width of seven samples fabricated by volunteer #7 operated by the robotic arm.

(Supplementary Fig. 9). The predicted values of $\beta$ by FEA under various magnetic fields up to 200 mT are validated by experimental measurements (Fig. 2b, c and Supplementary Fig. 10). Note that the rest angle equals 130° (i.e., $\beta_0 = 130°$) and when the magnetic field strength is 200 mT, the magneto-origami is completed folded (i.e., $\beta = 180°$). Upon removal of the actuation field, the magneto-origami machine rapidly recovers the rest state. This reversible folding and unfolding process is shown in Supplementary Movie 2.

Given that different origami structures require different folding procedures, the current fabrication of magneto-origami machines inevitably involves a manual folding process, despite that the 2D magnetic sheet pattern can be automatedly fabricated by the roll-to-roll process. To verify such a manual folding process has little influence on the magneto-origami machines, two statistical studies of eight volunteers each of whom folds seven samples

have been conducted. First, a single-crease magneto-origami machine is selected, and the average angle $\beta_0 = 130°$ with a standard deviation of 4° at rest state are consistently achieved (Supplementary Fig. 11). Second, a multi-crease Miura magneto-origami is selected owing to its complex folding process (Fig. 2d and Supplementary Movie 3). The freshly folded Miura magneto-origami have an average width of 2.4 mm with a very small standard deviation of 0.14 mm (Supplementary Fig. 11). Notably, such a folding process only takes about 140 s per sample, indicating high 3D fabrication efficiency of the origami technique. In addition, we also evaluate the deformed shapes of Miura magneto-origami machines folded by different volunteers in magnetic fields. We employ a robotic arm (Fig. 2e, myCobot Pro 600, Elephant Robotics Co., Ltd., China) to precisely control the actuation distance (glass plate to magnet center, denoted as $d$ in Fig. 2f). By varying $d$, the magnetic field strength can be

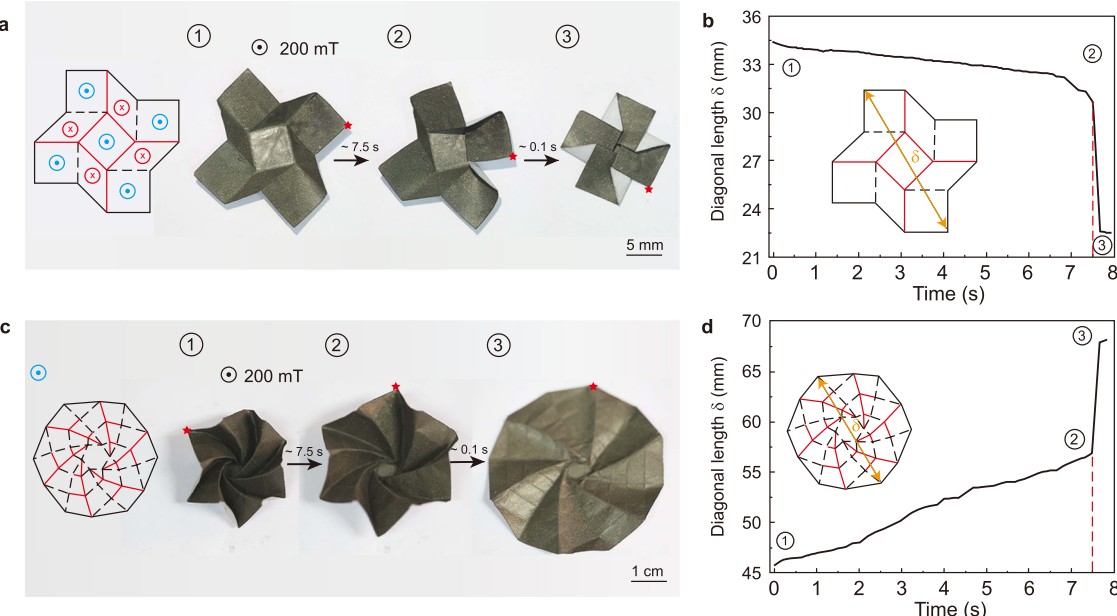

**Fig. 3 Folding/unfolding of deployable magneto-origami machines under a magnetic field of 200 mT. a** Twisting and folding process of the magneto-origami square–twist that is magnetized at the folded state. **b** Diagonal length $\delta$ of magneto-origami square–twist during the folding process. **c** The unfolding process of the magneto-origami starshade that is magnetized in the flat state. The red marker indicates the movement of a marginal point during the deployment process. The blue sign indicates a magnetization pointing outward. **d** The diagonal length $\delta$ of the magneto-origami starshade during the deployment process.

effectively tuned (e.g., B = 50 mT at $d$ = 43.0 mm, B = 100 mT at $d$ = 32.2 mm, Supplementary Fig. 8) to change the width of Miura magneto-origami machines (Fig. 2g, h and Supplementary Movie 4). Statistical studies only show a small variation between all samples in the deformed width and excellent reversibility when the magnetic field is removed (Supplementary Fig. 11), manifesting that the proposed fabrication method by roll-to-roll process and manual folding can construct magneto-origami machines with consistent geometries and shape-morphing performance.

It is also worth noting that the robot arm may only be able to complete regular modes of manipulation such as translation, circulation, and rotation (Supplementary Fig. 12). In this regard, manual operation by human hands may supplement some complex modes of manipulation. To verify the accuracy of manual operation, we compare both the measured magnetic field strength and deformed shapes of Miura magneto-origami machines operated by the robotic arm and human hands. Figure 2i shows that there is small difference in the measured magnetic field strength when tuning the actuation distance $d$ = 32.2 mm (B = 100 mT) and $d$ = 43.0 mm (B = 50 mT) by robotic arm and hands of eight volunteers. Besides, eight volunteers manually tuned the actuation distance and recorded the mean width of seven samples by themselves in Fig. 2j (solid curves). These results are very close to that by robotic arm manipulation (dashed black curve in Fig. 2j and Supplementary Fig. 11), suggesting that manual manipulation of Miura magneto-origami machines is also reliable.

**Folding/unfolding of deployable magneto-origami machines**. By changing the size from folded to unfolding state or vice versa, deployable structures show great promise in applications such as medical devices (e.g., vascular stents) and solar panels for spacecraft[38]. Here, we present two deployable magneto-origami machines that can morph into different configurations by applying a magnetic field in Fig. 3 and Supplementary Movie 5. The magnet is placed beneath the magneto-machine at a distance

of 22.5 mm (B ~200 mT). First, we present a magneto-origami square–twist in Fig. 3a. The magneto-origami square–twist consists of alternating square and rhombus facets with tailored mountain and valley folds. Being magnetized at the folded state, it has a fourfold rotational symmetry in the magnetic polarity patterns when unfolded. At the folded state, it is stable and one needs to apply a certain force to unfold it (Supplementary Fig. 13). The required force is further enhanced when the magneto-origami square–twist is in a magnetic field because the magnetic field provides additional magnetic forces preventing the unfolding. When it is unfolded by mechanical force, it can remain flat in the absence of a magnetic field. Applying the actuation magnetic field will trigger the snap transition from unfolded state to folded state as manifested by the abrupt change in the diagonal length $\delta$ in Fig. 3b. Next, a deployable magneto-origami starshade is developed in reminiscence of the starshade that blocks the glare of stars for the space telescopes (Fig. 3c). Different from the square–twist that is magnetized in its folded state, the magnetic sheet is first magnetized and then folded into a starshade (further discussions on this fabrication method are provided in the Discussion section). Under the magnetic actuation, the magneto-origami starshade gradually expands while rotating around its center (indicated by the red marker at the margin in Fig. 3c). After 7.5 s, the magneto-origami starshade quickly unfurls to a nearly flat sheet evidenced by the steep increase of its diagonal length $\delta$ (Fig. 3d). It is worth noting that both the folding of the magneto-origami square–twist and the unfolding of the magneto-origami starshades complete in a short time of 8 s with a change in the area by two folds. The large area change is intrinsically enabled by the origami structure and the on-demand shape-morphing is highly desired for applications such as energy harvesting[39,40].

**Sequential folding/unfolding of deployable magneto-origami machines**. The programmable sequence plays a key role in the self-assembly of 3D complex structures[41–43]. Here, we present

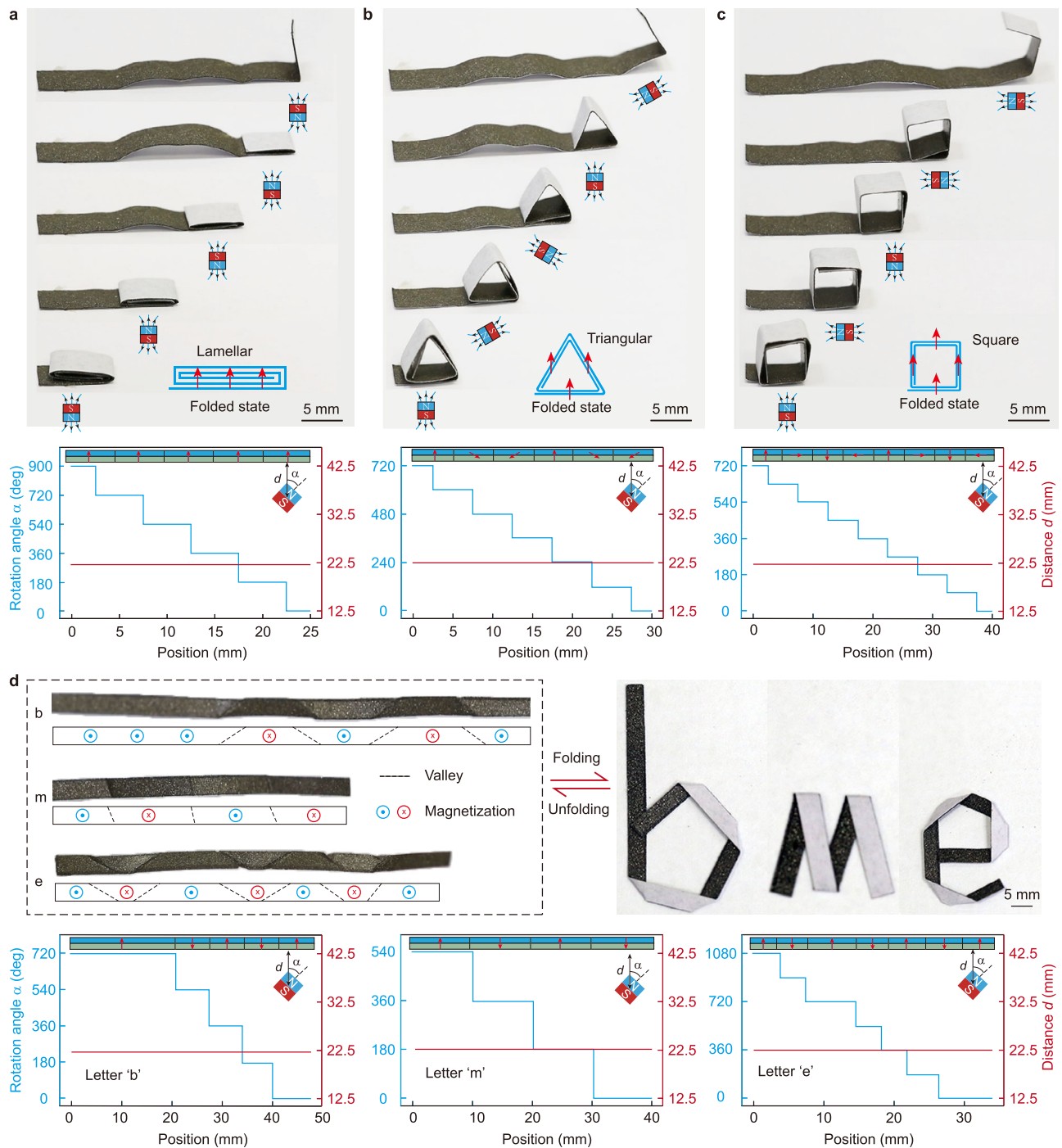

**Fig. 4 Sequential folding and unfolding of the magneto-origami strips. a–c** The schematic illustrations and experimental demonstrations of magneto-origami strips programmed with alternating magnetic polarity patterns. The magneto-origami strips can be sequentially folded into predesigned 3D shapes: **a** lamellar, **b** triangular, and **c** square, by moving and rotating the permanent magnet. Quantitative control schemes are provided in which rational angle α (left axis, blue curve) and distance d between the magnet and magneto-origami strip is plotted as a function of the magnet position. **d** Experimental demonstrations and quantitative control schemes of sequential folding of magneto-origami strips into letters "b", "m", and "e".

three tailored magneto-origami strips with sequential folding/unfolding ability via manipulating the actuation magnetic field. The 2D magnetic strip can be sequentially folded into a lamellar (Fig. 4a), triangle (Fig. 4b), and rectangle (Fig. 4c) origami, respectively. After being magnetized, they are imparted with alternating magnetization patterns. Different from the actuation mode in Fig. 3 where the magnet is placed at a fixed position during the entire deployment process, we move and rotate the magnet to realize the sequential folding and unfolding of the

magneto-origami strips. Corresponding quantitative control scheme by adjusting actuation distance d and rotation angle α of the magnet (see insert in Fig. 3a–c and Supplementary Fig. 8) while keeping the distance d = 22.5 mm (B ~200 mT) is provided for each strip. Note that such a sequential folding and unfolding is fully reversible by manipulating the magnet in the reverse direction. Taking the advantage of the sequential folding, we can encode different magnetization patterns into the magneto-origami strip. Here we demonstrate folding the magneto-

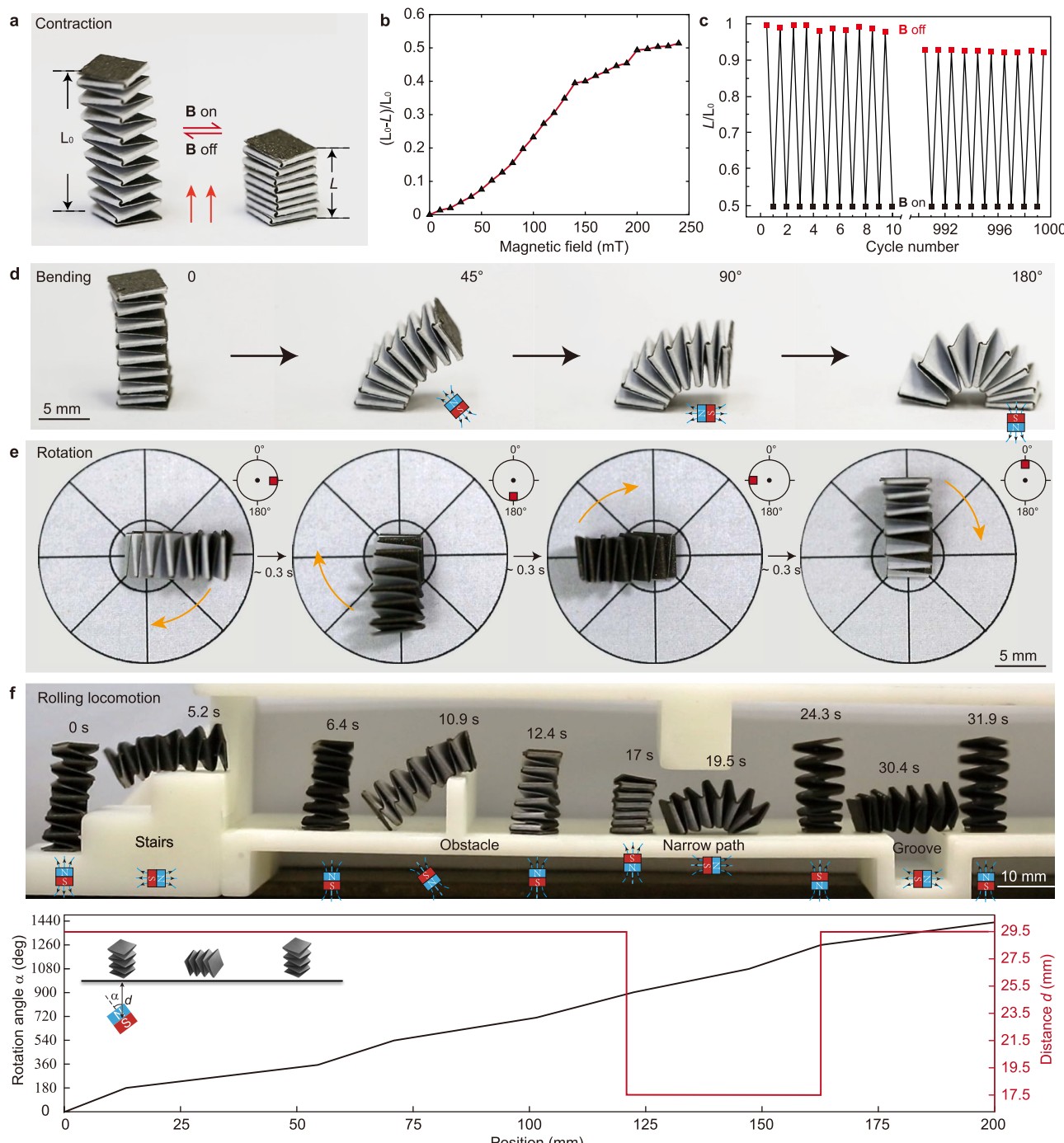

**Fig. 5 Actuations of the magneto-origami spring actuator. a** The spring actuator can contract when the magnet is placed in its axial direction. **b** The contraction ratio $(L_0 - L)/L_0$ of the spring actuator plotted as a function of magnetic field strength up to 220 mT. **c** The spring actuator remains 93% of the original length $L_0$ after a 1000-cycle actuation (B ~220 mT). **d** Bending behaviors of the spring actuator. Panels show bending angles are 0°, 45°, 90°, 180°, respectively. **e** The 360° rotation behavior of a bent spring actuator by rotating the magnet surrounding the actuator. The yellow arrow represents the rotation direction of the rotating magnet. **f** Experimental demonstration of the rolling locomotion of the spring actuator by passing through several obstructions including stairs, obstacles, narrow path, and groove. The height of the stair, obstacle, and narrow path is 10, 10, and 12 mm, respectively. The width of the groove is 10 mm. Quantitative control schemes are provided in which rational angle $\alpha$ (left axis, blue curve) and distance $d$ between the magnet and magneto-origami spring actuator is plotted as a function of the magnet position.

origami strips into letters "b", "m", and "e" in Fig. 4d. The corresponding origami and magnetization pattern of each magneto-origami strip is shown below the experimental pictures. A quantitative control scheme is also provided. The sequential folding and unfolding process is given in Supplementary Movie 6.

**The magneto-origami spring actuator.** Tuning the position and orientation of the cuboid magnet, magnet-origami machines can be actuated in different modes. Here, we present a magneto-origami spring actuator with diverse actuation modes in Fig. 5 and Supplementary Movie 7. First, the spring actuator can

contract when the magnet is placed in its axial direction (Fig. 5a). Moving the magnet toward the spring actuator, i.e., increasing the magnetic field strength, the spring actuator undergoes a contraction ratio (defined by $(L_0 - L)/L_0$) up to 50%, as shown in Fig. 5b. Upon removal of the magnet, it recovers the normal length $L_0$. Driven by the fast response of magnetic composite, the contraction and recovery process are realized within 0.3 and 0.25 s, respectively. Also, despite the multifold fabrication (Supplementary Fig. 14), this spring actuator shows remarkable repeatability by remaining 93% of its original length after 1000 cycles of actuation (Fig. 5c). Second, by fixing the base of the spring actuator, rotating the magnet around the spring actuator changes the magnetic field direction, yielding the bending configuration of the spring actuator, as shown in Fig. 5d. The bending angle can be precisely controlled up to 180° by manipulating the magnet orientation. Third, the bent spring actuator can further realize 360° rotation around its fixed end when rotating the magnet surrounding the actuator, as shown in Fig. 5e.

Last but not the least, the spring actuator can achieve rolling locomotion actuated by a rolling magnet. We demonstrate such rolling locomotion by navigating the spring actuator through several obstructions including stairs, obstacles, narrow path, and groove by controlling the distance $d$, rotation angle $\alpha$ and the position of the magnet in Fig. 5f (Supplementary Movie 7). First, rotating the magnet, the spring actuator can climb stairs and step over a 10 mm height obstacle easily. Next, by moving the magnet closer, the spring actuator is contracted from 18 mm to 11 mm, followed by a quick transit through a narrow path with a height of 12 mm. Thereafter, it recovers to its normal length and successfully crosses a groove with a width of 10 mm. Due to the excellent ability of shape-morphing and rolling locomotion, the spring actuator accomplished those complex tasks under untethered magnetic actuation in an enclosed environment around the 30 s. This demonstration suggests that the origami-enable large stretch ratio can effectively change the shape of the robot, making it adaptable to different tasks and environments. The moving speed of the spring actuator is about 2 mm s$^{-1}$ which is comparable to relevant studies (Supplementary Fig. 15).

**Magneto-origami charging robot**. Toward potential medical applications, we develop a deployable magneto-origami charging robot (MC robot) with remotely controlled locomotion and wireless charging capability (Fig. 6a–c and Supplementary Movie 8)[44]. On the paper side of the 2D sheet, the MC robot consists of an electronic circuit that can be wirelessly charged by an induction coil (Supplementary Fig. 16). The MC robot can be folded into a cuboid-like structure with dimensions of $16 \times 16 \times 10$ mm (Fig. 6a). We demonstrate the motion, deploying, and wireless charging in a pig stomach phantom. By vibrating the magnet at a frequency of 40 Hz, the MC robot can navigate across the uneven surfaces of the stomach phantom at a speed of ~1 mm/s. When it reaches the target area, on-demand unfolding is performed by moving the magnet closer to the MC robot (Fig. 6b). Thereafter, an alternative magnetic field is applied to generate an electric current, producing a stable voltage of 3.3 V that lights two LEDs (Fig. 6c). Note that LEDs are simplified representative models to demonstrate the wireless charging capability. They can be readily replaced by other functional electronics. This demonstration clearly shows that utilizing the 3D-to-2D shape-morphing, the magneto-origami machine can have on-demand large area change. By incorporating an electronic circuit, it is further empowered with wireless charging capability that can potentially provide for the long-term application of implantable electronic devices.

**Magneto-origami 8-3 encoder**. Analogous to digital devices, magnetic soft robots can also be programmed as mechanical encoders[45]. Utilizing the bending and rotation configurations of the spring actuator in Fig. 5, we present a magneto-origami 8-3 mechanical encoder in Fig. 6d–f. The logical circuit design of the 8-3 encoder is illustrated in Fig. 6d where Y0–Y7 represents eight input channels, and the on/off state of 3 LED indicates the output A0–A2 of the encoder. The electronic circuit design of the 8-3 encoder is shown in Supplementary Fig. 17 and the truth table of the encoder is given in Fig. 6e. To endow the conductivity of the encoder, the spring actuator is coated with a thin layer of gold. Initially, one end of the magneto-origami spring actuator is fixed at the center and connected with the positive pole of the power supply. Upon application of the magnetic field, the spring actuator bends and touches one input of Y0–Y7, turning on/off the corresponding LEDs. Four representative modes are shown in shown Fig. 6f and Supplementary Movie 9. For example, when the spring actuator contacts with Y0, all LEDs are turned off (i.e., A0 = A1 = A2 = 0); when the spring actuator contacts with Y3, LED0, and LED1 are turned on while LED2 is off (i.e., A0 = A1 = 1, A2 = 0).

**Magneto-origami quadruped robot**. Combining the deployable flower gripper and the spring actuator (Fig. 6), we develop a magneto-origami quadruped robot that can execute cargo-release tasks (Fig. 6g and Supplementary Movie 10). Four spring actuators are equipped as the legs of the robot for transportation during which the gripper can firmly grasp the cargo. The transportation of the cargo is enabled by actuating the two legs in the front. First, the frontal legs bend forward by moving the magnet forward. Then, manipulating the magnet a little bit away will cause the bent frontal legs to drag the robot forward. In a gait circle, the robot can stride with a displacement of 2 mm in less than 2 s (Fig. 6h). By repeating such a gait circle, the quadruped robot can walk forward step by step rapidly. When it reaches the destination, the magnet is moved closer to the flower gripper to open the petal such that the cargo is released on time (Supplementary Fig. 18). In comparison, a single-legged magneto-origami robot was also tested. It can carry the ball and move driven by magnetic attraction. But it fails when unloading the ball at the destination (Supplementary Fig. 19 and Movie 10). The demonstration of the quadruped robot manifests that by combining different magneto-origami, we can construct magnet soft robots with advanced functionalities for complex tasks.

**Magneto-origami arts/crafts**. Enabled by the intrinsic folding capability of the magnetic sheet, magneto-origami machines can be built into magnetically responsive arts/crafts. By printing colorful pictures on the top paper, we can further create magneto-origami arts/crafts. As a representative demonstration, we present a magneto-origami art—dancing butterfly and blooming flower—in Fig. 6i–k. The designs of the magneto-origami butterfly and flower are shown in Supplementary Fig. 20. By controlling the magnet via the motor, the magneto-origami butterfly flaps its wings while the flower is blooming. Background music can be played during the actuation process, leading to a beautiful and euphonic art (Supplementary Movie 11). We envisage that one can manually control the motor to trigger the shape-morphing of the magneto-origami machines, enjoying the fun of the hands-on operation and the beauty of the vivid art. Serving as a media for the dissemination of science and art, magneto-origami arts/crafts may have a far-reaching educational impact on public audiences, especially kids and students if exhibited in museums and schools.

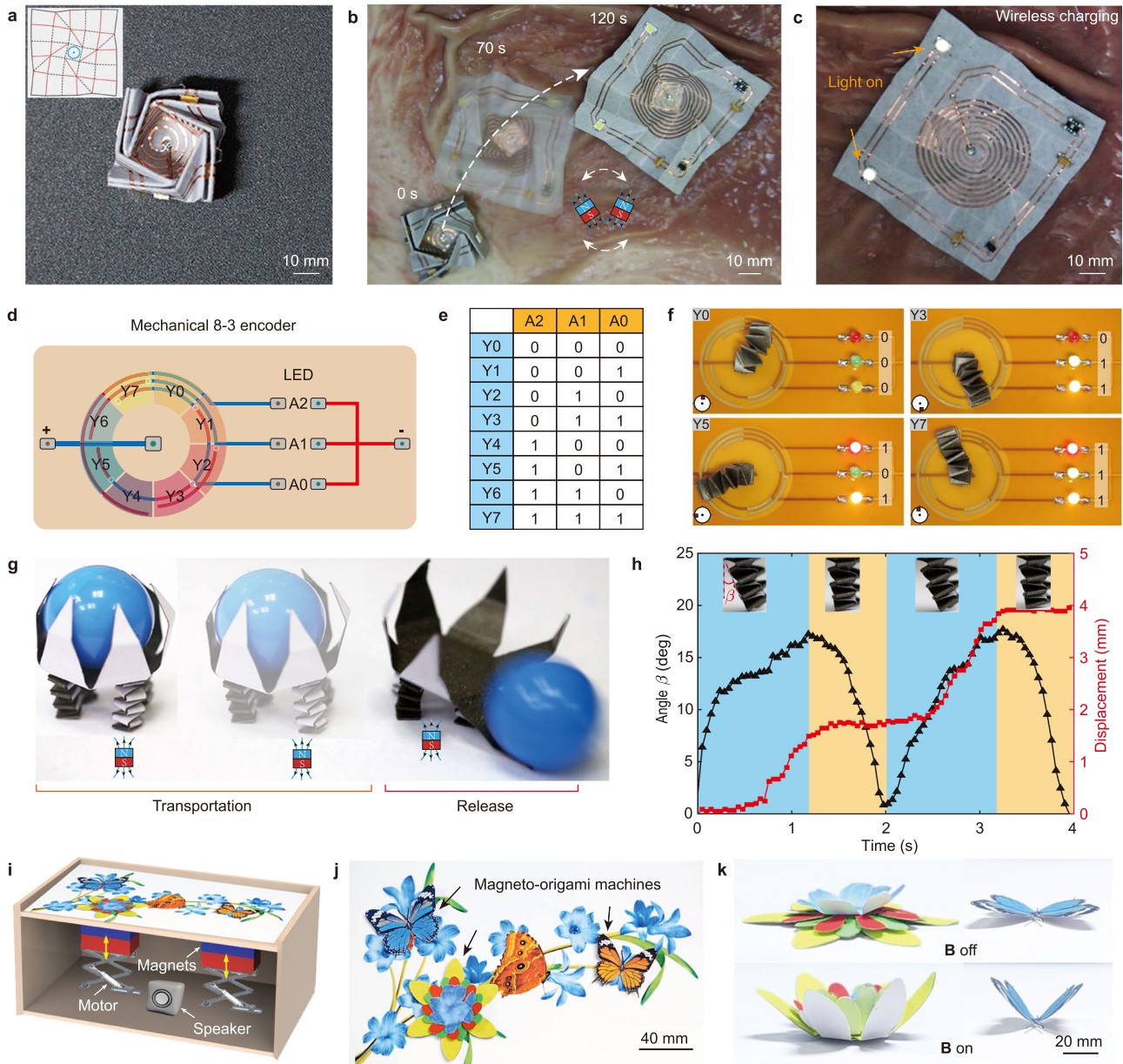

**Fig. 6 Demonstrations of applications of magneto-origami machines. a** The optical image of the folded magneto-origami charging robot (MC robot). Insert: the origami pattern of the MC robot. **b** The locomotion and on-demand deployment of the MC robot in a pig stomach phantom by vibrating the magnet at the frequency of 40 Hz. **c** The deployed MC robot can be remotely charged by the alternating magnetic field. **d** The logical circuit design of mechanical 8–3 encoder where Y0–Y7 represent the input channel and A0–A2 denote the output. **e** The truth table for the magneto-origami mechanical encoder. **f** Representative examples of the 8–3 mechanical encoder under different inputs. The red square in the circular inset represents the position of the magnet. **g** Dynamic locomotion and on-demand cargo release of the magneto-origami quadruped robot. **h** The bending angle (black line) and transportation displacement of the frontal legs (red line) in two actuation cycles. **i** By controlling the magnet via the motor, the butterfly can flap its wings and the flower is blooming up, while beautiful background music can be played during the actuation process. **j** Top view of the magneto-origami art. **k** Shape-morphing of the flower and butterfly.

## Discussion

The ancient art of paper folding, Origami, has been extensively explored as a fabrication strategy of engineering structures via a sequence of spatially organized folds. The key benefit of origami fabrication is the simple design principle and efficient creation of complex structures with customized geometries by constructing patterns in a plane and folding them into their final shape[25]. When utilized to design robots, origami robots exhibit intrinsic shape-morphing capability between 2D and 3D configurations. Enabled by multi-folds, the shape-morphing can have a large degree of deformation (e.g., stretching, contraction,

and bending), adapting the origami robots to different tasks and environments. In this work, we transform the conventional paper into a magnetic sheet by coating a thin layer of magnetic composite on the paper. The magnetic sheet preserves the paper-like foldability while imparting magnetic responsiveness to the magneto-origami machines it creates. We have built a roll-to-roll platform that achieves an automated fabrication of 2D magnetic patterns and demonstrates several 3D magneto-origami machines that can change their morphologies and execute locomotion tasks on demand, e.g., deploying, locking, sequential folding, and cargo transportation. In this work, the

failure rate is negligibly small for commonly seen origami patterns to perform simple motions.

Currently, magneto-origami machines are still in their infancy. Although substantial efforts have been made on designing folding patterns in literature, fabrication is still heavily associated with manual folding. In this regard, we have done two reproducibility tests to evaluate the finished shape and the shape-morphing behavior of the magneto-origami machine. Future work can be focused on how to develop the automated line for folding origami with consistent shape and performance. In addition, the magnetization of magneto-origami machines can also be encoded before they are folded, giving rise to different magnetic polarity patterns and shape-morphing behavior of the machine (Supplementary Fig. 21 and Supplementary Movie 12).

## Methods

**Preparation of magnetic composite**. The Ecoflex 00-10 was purchased from Smooth-On, Inc. NeFeB microparticles (average diameter: 30 µm) were obtained from Magnequench Co. Ltd. The composite ink was firstly prepared by mixing Ecoflex 00-10 with NdFeB microparticles with a particle volume fraction of 25 %, followed by sufficient stirring and degassing.

**Roll-to-roll fabrication of magnetic sheets**. The roll-to-roll scraping process is realized by a scraping coating machine (MSK-AFA-EC200, Shenzhen Kejing, China). The roll paper with the magnetic composite coating is scraped evenly with a scraper, and the distance between the scraper and the paper surface is set to 250 µm. The paper is heated by the dryer, the effective heating length of the dryer is 0.5 m, the temperature is set to 75 °C, the delivery speed of the paper is 100 mm min$^{-1}$ and the paint is dried after this step. The roll paper continues to go through the laser cutting machine (PWJDD-20W, Pinzan, China) for cutting according to the pre-designed pattern. After passing through each transfer roller, in turn, the cut roll is finally rolled on the collection roller.

**Magneto-origami machines**. The fabrication of the magneto-origami machines followed the same tearing, folding and magnetizing process. The patterns were designed using SolidWorks (Version 2016, Dassault Systems, France) and engraved via a laser cuter. The pattern was manually folded into a 3D shape and subsequently magnetized by an impulse magnetic field (3 T) using a magnetizer (MA-2030, Shenzhen JiuJu Company, China). For the mechanical 8-3 encoder, to make the spring actuator conductive, the magnetic paper was coated with an Au film by radiofrequency magnetron sputtering deposition (VTC300, Shenyang Micro Technology Co., Ltd, China). Then the conductive spring can be utilized as the connection in the logic system. The ME-robot was fabricated by transferring a layer of copper wire on the magnetic sheet. The copper wire was obtained by laser cutting a copper layer (thickness ~50 µm) and removing the needless part. The electronic components (rectifier, capacitance, voltage regulator) and the copper wire were joined by soldering.

**Characterization**. The surface topography of dual-layer magnetic paper was observed using an industrial digital camera (CM2000, KUY NICE, China). The magnetic flux density of magneto-origami was measured by a gauss meter (Shanghai Daxue Electromagnetic Equipment Co., Ltd., China). The contractions strain and folding angle of magneto-origami actuators were characterized by analyzing the Supplementary Movies recorded by a digital camera (EOS 5D, Canon, Japan) using Image J. The magnetic hysteresis loop of bilayer matter was obtained using a vibrating sample magnetometer (VSM, Lake Shore 7410, USA).

**Manipulation**. The spatially varying magnetic actuating field was generated by a NdFeB permanent magnet (N42, 50 × 25 × 25 mm, Beijing Zhongke Sanhuan High Technology Co., Ltd., China) for manipulation of magneto-origami machines. Dynamical transformations of magneto-origami machines were realized by the combination of vertical, horizontal, and rotational movements of the magnet. The magnetic field can be tuned via the distance between magneto-origami machines and the magnets.

**Statistics and reproducibility**. The statistical experiment (in Fig. 2) was performed by 8 adult volunteers, 8 males (ages: 29, 28, 24, 23, 25, 22, and 31) and 1 female (age: 26). No data were excluded from the analyses.

**Reporting summary**. Further information on research design is available in the Nature Research Reporting Summary linked to this article.

## Data availability

The authors declare that the main data supporting the findings of this study are available within the article and its Supplementary Information files. Extra data are available from the corresponding author upon request.

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

## Acknowledgements
This work was supported by the National Natural Science Foundation of China (No. 51975597), the Natural Science Foundation of Guangdong Province (Nos. 2019A1515011011 and 2021A1515011712), the General Program of Shenzhen Innovation Funding (No. JCYJ20170818164246179), the Special Support Plan for High Level Talents in Guangdong Province (No. 2017TQ04X674), the Fundamental Research Funds for the Central Universities (No. 20lgzd27), the Science Technology of the Shenzhen Sci-Tech Fund (No. KYTDPT20181011104007), the Guangdong Provincial Key Laboratory Program (No. 2021B1212040001), and the Science and Technology of Shenzhen Planning Foundation (Project No. RCBS20200714114922245).

## Author contributions
S.Y., C.F.G., Z.W., and L.J. designed the research. S.Y., L.W., J.W., P.L., Z.C., L.P., and X.S. performed the experiments. L.W., and Z.C. contributed to the mathematical model. Y.Z., L.W., and Q.L. performed the finite element analysis. S.Y., L.W., and Z.C wrote the paper. All authors approved the final paper.

## Competing interests
The authors declare no competing interests.
