## [Peer Review File · Nature Communications]

High-throughput fabrication of soft magneto-origami machinesEditorial Note: Parts of this Peer Review File have been redacted as indicated to remove third-party material where no permission to publish could be obtained.

Reviewers' comments:

Reviewer #1 (Remarks to the Author):

This manuscript presents a paradigm of creating origami machines actuated by magnetic forces. Several interesting examples are demonstrated. Origami-based compliant machines are attracting a lot of attention these years due to their rich design space and relatively simple kinematics. While the idea of magnetic origami is not new, this manuscript shows that such machines can be fabricated with rather cheap materials and simple procedures. However, the major issue for this manuscript is that the interpretations on the results are not deep and insightful enough. The manuscript is more like a technical report of a series of arbitrary trial and error. Therefore, the scientific value of this manuscript in its current form is low, and does not warrant publication on Nature Communications. In addition, some of the claims made by the authors are not well supported by the results shown.

With that being said, I think this work, once improved on the theoretical aspect, has the potential to become an impactful piece of research and be published on Nature Communications.

My detailed comments and suggestions are listed below:

1. There is a general lack of explanation and analysis on how to design and choose the origami patterns for the machines. For example, what are the geometric dimensions of the origami patterns? Are they rigid foldable? Can one predict their deformation before the experiments are performed? The authors may consider to add some quantitative theoretical and/or numerical analyses to supplement the experiments. Reference (21) listed some possible tools to be used.

2. It seems to me that although the magneto-origami requires easily accessible materials and equipment, the fabrication process still rely heavily on the skill of the person. For example, the composite sheet (paper + magnetized-elastomer sheet) must be first folded manually. It is not clear how this process would guarantee uniform outcomes. Typically, it is even difficult for a skilled origamist to fold a piece of paper into the same rest configuration (i.e. having the same amount of inelastic deformation), not mentioning that the mechanical properties of the composite sheet could become even more complicated than plain paper. On a related note, the mechanical properties (such as tensile modulus, yield strength, toughness, etc) of the composite sheet should be reported. Adding up the lack of theoretical analysis and in-depth interpretation, there is little evidence for the reader to be convinced that the proposed paradigm can be easily repeated, especially by other research groups. Therefore, the claim that the proposed paradigm is suitable for mass production is not well supported.

3. More details on the control scheme need to be provided. As I understand from the current manuscript, it seems that the control of the folding of these origami machines are quite manual. A person must hold a piece of magnet and adjust it as the person observing the shape changing of the origami machines. It is not reported quantitatively how the magnets should be posed and transported. I also hope that the authors can comment on how this control scheme could be automated in future development. This is another issue that reduces the credibility of the author's claim about mass production.

4. The phrase "2D-to-4D fabrication" in the title is confusing. In the literature, people use 4D fabrication/ printing to refer to a process of fabrication that involves the product being produced in a premature 3D form, and then changing shape over time to another 3D form in its service state. Hence the name means initial (3D shape + time) to new 3D shape. However, if 2D and 4D are put together, then they must refer to the same thing. Clearly, 2D refers to the geometry of the initial configuration of the origami machines (i.e. being flat), but 4D refers to a process, which causes a mismatch of subject and thus is inappropriate. The authors should call their approach either 4D fabrication, or 2D-to-3D fabrication, but not 2D-to-4D.

5. How is the failure rate in these experiments? For example, failure of fabrication, failure of control, etc. It is important to provide a whole picture of the research to the readers, which is important for a scientific report.

6. Page 3, line 49: the elasto-plastic property of paper is very complex, it is not convincing to me that the authors could reliably "harness the predesigned plastic folding."

7. Page 3, line 52: please be more specific on what do you mean by "a large degree of recovery."

8. Page 4, line 78: please be more quantitative when describing "the folded origami recovers partially to the rest state." What are the rest angles of the folding creases? How do you make sure that they can be repeatedly achieved in fabrication?

9. Page 6, line 124: the bistable snapping behavior of the square origami twist is an interesting intrinsic property arising from its geometry. It would be nice to see some simulations and analysis on how this property interacts with the magnetic forces applied onto the origami panels. Is the bistability enhanced by the magnetic forces? Or hindered but still preserved to some extent?

Reviewer #2 (Remarks to the Author):

1. General comments:

The manuscript introduces a new method of manufacturing sheets of magnetically-responsive materials that can be cut with origami patterns and externally actuated to make prescribed motions. The process could be extended to a roll-to-roll manufacture of functional remote-controlled robots. This is an interesting idea, and the authors used their new process to implement some impressive improvements on the state of the art of magnetically-actuated robots. I have some comments that I think would improve the paper, presented as general comments here and specific comments in subsequent sections denoted with double-hash marks ("##").

1. For completeness, please compare more quantitatively and directly with the most relevant prior art.

1.1 Most importantly, the authors should discuss reference 9 in more detail. H. Song et al. present a reprogrammable magnetically-actuated origami sheet.

1.2 Additionally, to refresh myself on this niche area (magnetically-responsive origami robots), I did a few quick search on Google Scholar with naive terms "magnet origami" and "magnetic origami"... and found several un-mentioned references that serve a similar purpose (small magnetically-actuated, possibly bio-compatible robots). While extensive introductions are not in the style of Nature Communications, I think - especially given the inclusion of Table S1 - more thorough description of prior art is warranted.

Daniella Rus has several papers in this area

[1] <https://ieeexplore.ieee.org/stamp/stamp.jsp?tp=&arnumber=7139386>

[2] <https://robotics.sciencemag.org/content/robotics/2/10/eaao4369.full.pdf>

[3] <https://ieeexplore.ieee.org/stamp/stamp.jsp?tp=&arnumber=7989560>

[4] <https://ieeexplore.ieee.org/stamp/stamp.jsp?tp=&arnumber=7487222>

Other authors

[5] <https://onlinelibrary.wiley.com/doi/epdf/10.1002/adfm.201904977>

[6] <https://pubs.rsc.org/en/content/articlehtml/2014/tc/c4tc00787e>

1.3 Finally, I think the comparison with other methods in Table S1 is over-simplified and unjustified, especially in light of its limited scope. Why did the authors omit reference 9 from this comparison? The current presentation makes it seem like the chosen competitors were hand-picked to create a favorable comparison, rather than dispassionately evaluate the current work in context of the state of the art.

2. The claims (and figures) about roll-to-roll manufacturing are unjustified and/or misleading. Please show more details about the specific process used to make the robots, and remove all schematics showing theoretically-possible processing steps (primarily referring to Fig. 1a), or at least make it clearer what was done vs. what could be done. This is especially important, since the manufacturing method is the main contribution of the manuscript. Make your contributions shine,

and avoid over-claiming. If you do this well, readers will be able to more readily learn from your work and integrate the introduced concepts into their studies --- which is the main value in scientific publications.

3. Several of the figures (such as 1a, 1d-e, 2 all, 3 all) are confusing, which detracts from the readability of the manuscript.

4. I would suggest doing multiple trials with multiple samples for most of the experiments, so confidence intervals may be obtained and readers can gain confidence in the repeatability of the manufacturing process. In absence of explicit discussion of repeatability problems, whenever one sample is shown, some readers will wonder whether the authors simply chose to present the only sample that worked.

5. The materials and methods section was lacking some important details. I commented on a few below, but I encourage the authors to attempt to repeat their experiments solely based on the materials and methods. Alternatively, ask a colleague to draw the whole process on a whiteboard based on the M&M section. This will highlight even more deficiencies than I showed.

Congratulations to the authors on their work so far, and I look forward to seeing their future work (including any revisions of this paper that I am sent). The manuscript is interesting, and I hope my comments can help improve the rigor and clarity of presentation.

2. Specific comments:

Introduction

- "mass and fast fabrication" -- this "mass" sounds like "kilograms" rather than "high-throughput" or "large-quantity"
- What makes these 4D? I have this criticism of most other studies' claims of 4D as well, but for consistency I must bring it up. This seems like 2D-3D deployment. Please justify the choice of this 4D term.
- I recommend doing a more in-depth comparison to other magnetically-actuated origami robots, to clarify the contribution further.

Results and Discussion

- "self-folding" is inapplicable here. The films are actuated externally. In order to fold the structures multiple times, you need an external 3-DOF system to {lift, move, and rotate} the external magnet.
- (minor) "magneto-origami electronic robot" - What does that mean? What makes this robot electronic, in contrast with other magnetically-actuated sheets (such as those referenced as refs. 1, 3, 9, 13, 14)?

Materials and Methods

- "Finally, a bilayer magnetic paper was fabricated." -- how?
- Why was the spring coated with Au? Is this for making connections in the logic device? I'd recommend explicitly stating this
- How were the magnets moved for the various examples shown in the paper? By hand? With a robot arm? With a multi-DOF gantry/stage?

Supplementary materials (mostly small comments)

- How was the magnetic field in Figure S5 measured?
- Fig S7 - b x-axis should be "cycles". It might be helpful to calculate the mean and standard deviation of the peak and valley (max and min) angles.
- Fig S8 is confusing. Can you mark the start, and explain what the squares are? Also, why did it go from 5x5 L-shape, down to 3x3 L-shape? Why not go to a single 1-square shape for maximum springiness? In the videos and other figures, it seems you went to a 1-square shape.
- Fig. S9: I think "voltage regulator" is a more readily-used term than "voltage stabilizer". Also, what's the point of the wireless charging (also relates to Fig. 5b), if there are not onboard

batteries? If the point of this is to show that circuits can be embedded onto the ME-robot, then I think vibrating in a pig stomach phantom and activating a light is a little strange of a demonstration. Something more natural for robotics might be adding a sensor suite, or a microcontroller plus some actuators and/or sensors. This is not a technical criticism, just a thought on the presentation / appeal to wider audiences.

3. Comments on figures:

Fig 1:

- (major) 1a Please be precise about the claims of roll-to-roll capabilities.
- The shown schematic does not look like roll-to-roll.
- I suggest including a photo of the actual setup (maybe in the supplemental).
- Please provide evidence of the roll-to-roll process. I'm skeptical that the scraping, curing, laser, and tearing off all happen in a roll-roll process. Thus, I recommend reframing a lot of the paper and schematics to more clearly show the actual, implemented process --- not what could be done in several months/years "if XYZ improvements happened".
- (minor) typo: "tunning"  "tuning"
- (minor) Why is D a 4-row Miura structure, while e is 5? Style choice, it would be simpler to interpret with the same sample for both subfigures.
- (major) I was left wondering what I was even seeing in d. d has paper with colored cut lines, while e is just the black elastomer. By the end of C, I assumed the paper is just the back side, while the other side is black elastomer. Then the figure d,e images and caption added more confusion. Did you tape on some paper for sample in d row by row, while e you skipped taping paper? Suggestions: can you show the same sample at different stages of manufacturing, starting from c onward? Maybe even some images of the roll-to-roll process would help. Maybe being more descriptive in the captions could help.

Fig. 2:

- The magnetization signs above subfigures a-e are all confusing, rendering the figure very difficult to interpret.
- For the images, I'd just put text showing the current magnetization level. Unless the magnetization is turned from 0 to 200 mT instantaneously at $t=0$, in which case I don't even think it's necessary to show b on vs off. I'd just label when it's turned off.
- Also, if the magnetization off yielded a sudden jump in diagonal length, this leaves me wondering... how were the initial conditions defined? Why didn't they start with magnetization off, then add magnetization for N seconds, then turn magnetization off? I think that would be more clear.
- Why does the magnetization appear to start halfway through the first image?
- For consistency (if you choose to keep the magnetization bars in their present condition), Fig. c should have the "B off" on the far right after 7.5 s.
- For consistency, c and e should go to the same time.
- Why does the base pattern for 2a have four petals, while the images have >10 ?

Fig. 3:

- (minor) The "letters" start as strips, then fold into letters. I'd put the letters to the right of the strips, instead of to the left.
- (minor) Similar comment on a-c: I'd recommend putting the chronologically-first image at the top-left, and have chronology go down for a-d (and right for d). I assume this chronology based on the figures as well as the accompanying supplemental videos.
- (minor) Please avoid tiny text. For example, the XYZ axes are super tiny. Readers will need to zoom in just to read the text. For super-small text, either make it larger if it's important, or delete if it's not important. For example, a-c the xyz axis doesn't add any information - you don't use xyz coordinates anywhere else when you discuss this figure.

Fig. 4:

- (minor) 4a: again... why bother adding an xyz axis here?
- (minor) 4c "circle number"  "cycle number"
- (minor) 4e cool, but the text and lines are distracting
- 4f - great subfigure. I might suggest adding the magnet orientations, similar to the videos. Try it; it might be distracting though.

Fig. 5:

- (minor) 4h: "...displacement of the frontal legs (red line) over two actuation cycles"

4. Comments on Multimedia (Videos, etc.)

- (minor) The title slide has tons of words and is super difficult to read in the time given. I would suggest removing the affiliations, shrinking the author list, simplifying the background, moving the movie titles (such as "Movie S4: Sequential folding and unfolding of the magneto-strips") up to the top and larger.

- (minor) Transitions between video segments are rather abrupt. Can you put fades, or transition slides?

- (major) Video 8 - Does the robot even need a 4-legged configuration? Try this demonstration with a single larger spring to support the gripper. It appears to just be bouncing and moving to stay above the magnet (kind of like the kids toys that you drag a magnet under a steel ball to solve a maze)

Reviewer #3 (Remarks to the Author):

The paper presents several interesting demonstrations of magneto responsive small robots based on different origami designs. However, compare to the recent advances in the relevant researches, novelty and contribution in the current manuscript are not convincing. Major concerns are as follows:

1. Although the wording is different, there exist similar mechanisms on soft material by utilizing magnetic field-driven small robots capable of locomotion and shape morphing, e.g. paper-based, polymer-based, and so on. In there, the authors can find easy and convenient methods of attaching magneto-responsive polymer to a paper or magneto-responsive polymer itself to achieve 4D (in author's wording) fabrication. Moreover, these researches used both B-field and gradient fields altogether to perform the required motion based on a simple origami robot configuration. Also, the origami designs in this study are simple and some are already available in the literature. On these aspects, the novelty and contribution are weak, and might not be advanced compare to the state-of-the-art technology.

2. The mechanism of magnet actuation is not depicted both in the experimental setup and supplementary video clip. Since the magnetic field intensity and gradient change with respect to the distance and magnetization direction, a detailed explanation of how the authors control the external magnetic field precisely (even with frequency) should be included. Moreover, the proposed mechanism should consider the effect of gradient field generated by a magnet for proper validation.

3. Can the spring actuator bend the shape without fixing the base (end of one side) on the floor? Maybe not. When the magnet moves as shown in Fig.4d, the spring actuator will be pulled to the magnet and will not make this motion. Detail explanation of the experimental condition for each example should be shown for proper evaluation of the methods.

4. What are the benefit or advantages of each demonstration in this study? In the reviewer's view, most of the demonstration can be achieved without the help of origami or like this complex design. A scientific or technological contribution should be addressed.

5. Quantification analysis (e.g. moving speed, magnetization property) compared to the relevant studies are requested for performance validation.

Reviewer #4 (Remarks to the Author):

This paper presents various performances of magnetically foldable and actuatable origami devices. The performances include actuation of simple deploying/folding structures, centimeter-sized shape-changing locomoting robots, and an electronics-equipped power transmitting origami sheet. Authors claim that due to the realization of "mass production method of the magneto-active machine(s) with customized architectures and advanced functionality", whose process is called "2D-to-4D fabrication" presented in the paper, the approach shows advantages to other fabrication methods such as laser programming, 2D molding (I believe this is what the authors intend to say in Table 2), 3D printing, and assembly-based fabrication. This is an effort-intensive study

presenting unique and reliable results. Many results are well achieved. I believe, however, with a few more tweaks detailed below and the paper could get better.

First, the technical challenge addressed in the paper is obscurely defined. If the main contribution is the method of automated production of magneto-active machines, the authors should more clearly show the limitations of the method with more clear presentations in the results and outcomes. I'd like to see a movie of what is presented in Figure 1 if the process is configured as depicted; e.g. the curing and the laser cutting are subsequently done. Also as this is a self-folding/unfolding study, what kind of shapes are self-foldable (can be automated in the fabrication) and not self-foldable with the proposed method, and more clear descriptions on the involvement of (human) processes in the fabrication and the control are required. Why can't the spring structure in Figure 4 or the quadruped robot in Figure 5g be self-folded, whereas sequential folding is presented as one of the method's abilities? Currently, these limitation boundaries are obscurely presented or not referenced at all, making grasping the novelty of the work difficult. I see also in the video that the processes of making crease patterns and initially folding the origami structures for magnetization are conducted manually by a human, and are not yet automated. It also looks like that the application of a 3T magnetic field is manually conducted. As they generally require complicated manual handling, I cannot wipe out the impression that the automation process of the study is still midway through. Authors may want to clarify the advancement of the process to the literature work.

Second, many of these performances such as deployable structures, sequential folding structures, a locomoting robot, wireless powering sheet, lack comparisons on their performances with the other works. Only two (review) papers on origami robots were referenced (line 48, refs 20 and 21) in this regard, making the comparison of results with other similar robotics performances difficult. For example, the authors claim that "The deployable magneto-origami electronic robot can unfurl by 12 times of its folded area..(line 218)". The achievement of 12 times is owing to the crease pattern invented by another person, and the impact of 12 times is difficult to assess in the context. While it is meaningful to see that the proposed method can be applied to different performances and applications, the presentation style of results gives an impression that the performances were heuristically sought and designed, and are not necessarily coherent. Limited use of passive voices in the text with a clearer presentation of the subjects should be considered (e.g. line 73: .. the torn pattern is folded into a desired 3D origami.., line 110 .. magneto-origami flower is constructed..; line 229 .. ink was uniformly coated..).

Given that the basic principle of "Programming magnetic anisotropy in polymeric microactuators" by Kim et al., <https://www.nature.com/articles/nmat3090> (2011) (which is missing in the reference) which was further advanced in (13) or (17), and therefore a contribution of the work is limited to the semi-automated fabrication method and the versatile applicability of the method in different performances, higher theoretical solidity in these aspects are wanted.

Minor comments appear below:

Line 54: I'm not sure if reference 27 should appear here in the context of origami structure.

Line 75: application of impulsive magnetic field for the wireless reprogramming of a magnetic (NdFeB) structure was presented by Sitti et al. "Remotely Addressable Magnetic Composite Micropumps. RSC Advances, 2012".

Authors often use the words "direct 2D-to-4D fabrication". What do you mean by "direct" here?

Line 94: ".. unfolding process is shown in..". The crease is only partially unfolded (and folded).

How does the motion of the cuboid magnet affect the folding angle?

Figure 3b and Figure 3c: It is not clear in the text how the magnet was moved to form different shapes. Can you mathematically formulate in the supporting material?

Line 170: ".. we develop a deployable magneto-origami electronic robot.." There are planar origami structures that can be empowered with magnetic induction (e.g. Rob Wood from Harvard microrobotics lab), even the same setup may not exist, some references to the previous research are needed.

Line 176: The authors claim that they demonstrated "electrical stimulation therapy". However, I don't see the stimulation performance. I assume the alternative magnetic field for magnetic induction was managed by a low-inductance coil which is not described in the paper.

Responses to Comments on “NCOMMS-21-29338”

We sincerely thank the editor and the four reviewers for thoroughly reading our manuscript. The constructive comments and suggestions are fully addressed point-by-point with key responses underlined in this letter. For the convenience of the reviewers, we marked the corresponding modifications in our revised manuscript in blue. We believe that the questions raised by the editor and all reviewers have highlighted areas in need of further attention and the modifications have contributed to a significant improvement of our manuscript, making it more suitable for publication in *Nature Communications*.

Reviewer # 1

General Comment. This manuscript presents a paradigm of creating origami machines actuated by magnetic forces. Several interesting examples are demonstrated. Origami-based compliant machines are attracting a lot of attention these years due to their rich design space and relatively simple kinematics. While the idea of magnetic origami is not new, this manuscript shows that such machines can be fabricated with rather cheap materials and simple procedures. However, the major issue for this manuscript is that the interpretations on the results are not deep and insightful enough. The manuscript is more like a technical report of a series of arbitrary trial and error. Therefore, the scientific value of this manuscript in its current form is low, and does not warrant publication on Nature Communications. In addition, some of the claims made by the authors are not well supported by the results shown.

With that being said, I think this work, once improved on the theoretical aspect, has the potential to become an impactful piece of research and be published on Nature Communications.

Responses. We thank the reviewer for your evaluation and comment on the potential impact of our work. We have significantly revised our work by adding theoretical and numerical analyses and discussions. Detailed revisions are provided below.

Comment 1.

There is a general lack of explanation and analysis on how to design and choose the origami patterns for the machines. For example, what are the geometric dimensions of the origami patterns? Are they rigid foldable? Can one predict their deformation before the experiments are performed? The authors may consider to add some quantitative theoretical and/or numerical analyses to supplement the experiments. Reference (21) listed some possible tools to be used.

Response 1.

In this work, we empower the traditional paper origami with magnetically responsive functionality by simply replacing the raw paper with a paper-based magnetic sheet (revised **Figure 1a**). The principle of designing magneto-origami patterns remains the same as that of traditional paper origami. As long as we know how to make paper origami, we can create the corresponding magneto-origami machine. The geometric dimension of the current origami patterns is in centimeters, and it can be easily scaled up. A broad range of magneto-origami machines with 3D geometries that we normally see in our daily life are showcased in the revised **Figure 1g**.

To further strengthen our work, we have built a roll-to-roll platform to achieve an automated fabrication of magnetic sheets with desired patterns as shown in the revised **Figure 1b-1e**.

Revised Figure 1. Fabrication of magneto-origami machines. (a) Schematic illustration of fabrication processes of the patterned magnetic sheet by roll-to-roll method, folding/magnetizing, and partial recovery. (b)-(e) Experimental pictures of (b) laser cutting, (c) scraper, (d) pattern, and (e) curing heater of the roll-to-roll platform to fabricate patterned magnetic sheets. (f) Images showing a roll of raw paper and magnetic sheet with a 2D origami pattern. (g) Various magneto-origami machines with customized geometries and high structural resolution can be consistently fabricated.

On Page 4, we add “**Fabrication of soft magneto-origami machines.** The magnetic composite consists of magnetizable neodymium-iron-boron (NdFeB) particles (average diameter $\sim 30 \mu\text{m}$) embedded in the polymer resin (i.e., Ecoflex 00-10, Smooth-On) with a particle volume fraction of 25%. The 2D magnetic sheet is made by coating and curing a layer of magnetic composite on a piece of raw paper (Figure 1a, Supplementary Figure 1). To achieve an automated fabrication of 2D magnetic sheets, we build a roll-to-roll platform as shown in Figure 1b-1e. The major components of roll-to-roll platform consist of a roller of raw paper, a scraper (Figure 1c) that creates an even layer of uncured magnetic composite on the raw paper, a heater (Figure 1e) that cures the magnetic composite layer, a laser machine (Figure 1d) that cuts the magnetic sheet into designed patterns, and an end roller. After curing, the designed 2D pattern is cut by a laser machine with uncut connections to allow for an easy tear-off (Supplementary Figure 1, Supplementary Movie 1). The rolling velocity, the temperature of the heater, and the pattern of the magnetic sheet in the roll-to-roll process can be controlled in an automated manner, which lays the foundation for the high-throughput fabrication of magneto-origami machines (see Materials and Methods). Supplementary Movie 1 shows the detailed roll-to-roll process in which the production rate is 100 mm/min.”

The folded crease of magneto-origami behaves like an elastic hinge that can be treated as rigid foldable. The elastic torque density has been experimentally measured (Supplementary Figure 6), which will be later used for predicting the shape-morphing behavior of magneto-origami machines in the numerical analysis.

Supplementary Figure 6. Measurement of the elastic torque constant of the folding crease.

On Page 5, we add “Notably, the folded magnetic sheet behaves like an elastic hinge (Supplementary Figure 6) and shows negligible change in the bending angle after 1000 cycles of folding and unfolding (Supplementary Figure 7). Such an excellent folding/unfolding reversibility makes it promising candidate for creating magneto-origami machines that can be actuated multiple times with consistent performance.”

To elucidate the shape-morphing mechanism of the magneto-origami machines, we provide some theoretical discussions and numerical analysis in the revised manuscript. We updated **Figure 2** and added **Supplementary Figures 8-9** in the revised manuscript.

Revised Figure 2. Theoretical analysis, finite element simulations, and reproducibility test of magneto-origami machines. (a) Schematic illustration showing the shape-morphing of the single-crease magneto-origami actuated by a magnetic field. Equilibrium is reached when the total driving magnetic torque T_m is equal to elastic torque T_e . (b) Comparison of deformation of single-crease origami between experiments and finite element simulations. (c) Rotation angle β as a function of applied magnetic field strength up to 200 mT. (d) Example of folding Miura magneto-origami from the 2D magnetic sheet, in which red solid lines represent the mountain folds while black dashed lines denote valley folds. The left insert shows the magnetic polarity pattern where cross means magnetization pointing inward while blue represents magnetization pointing outward. The right insert shows the finalized Miura magneto-origami. (e) Demonstration of shape morphing of Miura magneto-origami under different magnetic fields. (f) Statistical analysis of reproducibility and performance evaluation of Miura magneto-origami fabricated by 8 volunteers each of whom folded 7 samples.

Supplementary Figure 8. A cuboid permanent magnet for actuating magneto-origami machines. (a) Schematic of the cuboid magnet with 25 mm in width, 50 mm in length, and 25 mm in height. (b) The magnetic field around the cuboid magnet. The distribution and magnetization of the permanent magnet was obtained using COMSOL Multiphysics software, under the same conditions including dimensions and magnetization. (c) Schematic of two parameters (distance d and angle α) to quantitatively control the permanent magnet. (d) The magnetic field strength as a function of distance d along the centerline of the magnet ($\alpha = 0^\circ$).

Supplementary Figure 9. Finite element analysis (FEA) of a single-crease magneto-origami machine actuated by a cuboid permanent magnet. (a) FEA model of a single-crease magneto-origami machine. (b) The FEA model for single-crease magneto-origami machine and detailed parameters of the model. (c) Meshed FEA model in COMSOL. (d) Folding state of the single-fold model when the residual magnetization of the permanent magnet is $1000 \text{ kA} \cdot \text{m}^{-1}$.

On Page 7, we add “**Shape-morphing mechanism of magneto-origami machines.** Fully magnetized magneto-origami machines retain the residual magnetic polarity patterns, i.e., magnetization \mathbf{M} . When subjected to an actuation magnetic field, magneto-origami machines can deform and move. In this work, we select a cuboid permanent magnet (NdFeB, dimension: 50 mm × 25 mm × 20 mm, residual magnetization 103 kA/m) as the magnetic actuation source (Supplementary Figure 8) because it is easier to handle and can generate stronger magnetic fields (up to 200 mT in this work) than electromagnetic coils. By varying the position/orientation of the cuboid magnet, the magnetic field strength \mathbf{B} around the magneto-origami machine can be effectively changed (Supplementary Figure 8). For example, the magnetic field strength on the centerline of the cuboid magnet can reach up to 200 mT when the distance between the center of the magnet and the magneto-origami machine is around 22.5 mm (Supplementary Figure 8)

To elucidate the magnetically controllable shape-morphing mechanism, here we adopt a single-crease magneto-origami machine as a representative example (Figure 2a). Upon actuation, the upper panel of the magneto-origami machine rotates around the crease due to the driving force, i.e., the magnetic torque density $\mathbf{M} \times \mathbf{B}$ and body force $(\nabla \mathbf{B})\mathbf{M}$ where operator “ \times ” and “ ∇ ” represents the cross-product and gradient, respectively. Therefore, the total magnetic torque that deforms the upper panel can be calculated as

$$\mathbf{T}_m = \int_{\Omega} [\mathbf{M} \times \mathbf{B} + \mathbf{r} \times (\nabla \mathbf{B})\mathbf{M}] dv \quad (1)$$

where Ω represents the 3D domain of the upper panel and \mathbf{r} is the position vector of a spatial point on the upper panel with respect to the crease (Figure 2a). According to the literature² and our experimental validation (Supplementary Figure 6), the folding crease behaves like an elastic hinge in which the total elastic torque is linearly proportional to angle change of the crease, i.e.,

$$\mathbf{T}_e = kL(\beta - \beta_0) \quad (2)$$

where $k = 1.47 \mu\text{N}/\text{deg}$ represents the experimentally measured torsional constant of the crease per unit width, L , $\beta_0 \approx 130^\circ$ and β denote the width, the rest angle, and folding angle of single-crease magneto-origami, respectively (Figure 2a, Supplementary Figure 6). Therefore, the deformed shape of the magneto-origami machine can be found when the equilibrium is reached, i.e.,

$$\mathbf{T}_m = \mathbf{T}_e \quad (3)$$

Considering that magnetic fields around a cuboid magnet are nonuniform, Eq. (3) is solved by finite element analysis (FEA) (Supplementary Figure 9). The predicted folding angle β by FEA under various magnetic fields up to 200 mT are validated by experimental measurements (Figure 2b, 2c). Note that when the magnetic field strength is 200 mT, the magneto-origami is completely folded. Upon removal of the actuation field, the magneto-origami machine rapidly recovers the rest state. This reversible folding and unfolding process is shown in Movie 2.”

Comment 2.

It seems to me that although the magneto-origami requires easily accessible materials and equipment, the fabrication process still relies heavily on the skill of the person. For example, the composite sheet (paper + magnetized-elastomer sheet) must be first folded manually. It is not clear how this process would guarantee uniform outcomes. Typically, it is even difficult for a skilled origamist to fold a piece of paper into the same

rest configuration (i.e. having the same amount of inelastic deformation), not mentioning that the mechanical properties of the composite sheet could become even more complicated than plain paper. On a related note, the mechanical properties (such as tensile modulus, yield strength, toughness, etc) of the composite sheet should be reported. Adding up the lack of theoretical analysis and in-depth interpretation, there is little evidence for the reader to be convinced that the proposed paradigm can be easily repeated, especially by other research groups. Therefore, the claim that the proposed paradigm is suitable for mass production is not well supported.

Response 2.

Given that different designs of origami require different folding procedures, manual folding is seemingly inevitable currently. To make consistent foldings, each fold is first completed folded to be flat and then recover to the rest state as shown in revised **Figure 1a**. Here, we choose both a single-crease magneto-origami and multi-crease Miura magneto-origami as representative examples. To show that such a manual folding process has little influence on the magneto-origami machines, statistical studies with 8 volunteers each of whom folds 7 samples have been conducted (revised Figure 2d-2f). Statistics in Supplementary Figure 10 show that both the rest angle of the single-crease magneto-origami and width of Miura magneto-origami at rest state are almost the same folded by different volunteers. In addition, we also evaluate the shape-morphing performance of the Miura magneto-origami when actuated by a magnetic field. Results in revised Figure 2f clearly indicate a small variation between different samples.

On Page 7, we add “Given that different origami structures require different folding procedures, the current fabrication of magneto-origami machines inevitably involves a manual folding process, despite that the 2D magnetic sheet pattern can be automatically fabricated by the roll-to-roll process. To verify that such a manual folding process has little influence on the magneto-origami machines, two statistical studies of 8 volunteers each of whom folds 7 samples have been conducted. First, a single-crease magneto-origami machine is selected, and the average angle $\beta_0 = 130^\circ$ with a standard deviation of 4° at rest state are consistently achieved (Supplementary Figure 11a-b). Second, a multi-crease Miura magneto-origami is selected owing to its complex folding process (Figure 2d, Supplementary Figure 11 and Supplementary Movie 3). The freshly folded Miura magneto-origami have an average width of 2.4 mm with a very small standard deviation of 0.14 mm. Notably, such a folding process only takes about 140 s per sample, indicating high 3D fabrication efficiency of origami technique. In addition, when actuated in a magnetic field up to 200 mT (Figure 2e), they exhibit small differences in the deformed width (Figure 2f) and excellent reversibility when the magnetic field is removed (Supplementary Movie 3). These statistical studies manifest that the proposed fabrication method by roll-to-roll process and manual folding can construct magneto-origami machines with consistent geometries that enable controllable shape-morphing under magnetic actuation”

Supplementary Figure 11. Statistical studies of magneto-origami machines. (a) Optical image of a single-crease magneto-origami at the rest state. (b) The average and standard deviation of the rest angle of the single-crease magneto-origami is 130° and 4° , respectively. (c) Optical image of a Miura magneto-origami machine. (b) The average and standard deviation of the length of all the samples is approximately 2.4 mm and 0.14 mm, respectively. Such a folding process only takes about 140 s.

Mechanical properties of the magnetic sheet have been measured and reported in the revised Supplementary Figure 5.

On Page 5, we add “By coating a $100\ \mu\text{m}$ -thick magnetic composite layer on top of a $90\ \mu\text{m}$ -thick raw paper (Figure 1f, Supplementary Figure 4), the paper-based magnetic sheet (Young’s modulus $\sim 7\ \text{MPa}$, ultimate tensile stress $\sim 14\ \text{MPa}$, Supplementary Figure 5) preserves the excellent folding capability of paper, allowing for consistent fabrication of magneto-origami machines in a broad spectrum of geometries with a high structural resolution, as showcased in **Figure 1g**”

Supplementary Figure 5. Mechanical properties of the magnetic sheet measured by tensile test.

Considering the built roll-to-roll fabrication of magnetic sheets in **Response 1** and the theoretical/numerical analysis in **Response 2**, we believe our current approach is capable of fabricating magneto-origami machines in a large scale without noticeable variation in both finished shape and performance.

Comment 3.

More details on the control scheme need to be provided. As I understand from the current manuscript, it seems that the control of the folding of these origami machines are quite manual. A person must hold a piece of magnet and adjust it as the person observing the shape changing of the origami machines. It is not reported quantitatively how the magnets should be posed and transported. I also hope that the authors can comment on how this control scheme could be automated in future development. This is another issue that reduces the credibility of the author's claim about mass production.

Response 3.

The cuboid magnet is selected as the magnetic source to control the magneto-origami machines because (1) compared with electromagnets, the magnetic field strength of cuboid magnet can be larger (up to 200 mT in our work); (2) handling/controlling a cuboid magnet is simple and has been widely used in literature. By adjusting the position/orientation of the magnet (Supplementary Figure 8c), the actuating magnetic field can be effectively tuned. To alleviate the concern of the reviewer, quantitative control schemes are provided for each demonstration in the revised Figure 4,5.

Although currently the magnet is controlled manually, we believe it can be readily integrated into any digital control platform such as a robotic arm. To alleviate concerns, we discussed our limitations and future perspectives in the revised manuscript.

On Page 7, we add “In this work, we select a cuboid permanent magnet (NdFeB, dimension: 50 mm × 25 mm × 20 mm, residual magnetization 10^3 kA/m) as the magnetic actuation source (Supplementary Figure 8a) because it is easy to handle and can generate stronger magnetic fields (up to 200 mT in this work) than electromagnetic coils. By varying the position/orientation of the cuboid magnet, the magnetic field strength B around the magneto-origami machine can be effectively changed (Supplementary Figure 8c). For example, the magnetic field strength on the centerline of the cuboid magnet can reach up to 200 mT when the distance between the center of the magnet and the magneto-origami machine is around 22.5 mm (Supplementary Figure 8d)”

On Page 9, we add “Corresponding quantitative control scheme by adjusting the position and rotation angle α of the magnet (see insert in Figure 3a-3c, Supplementary Figure 8c) while keeping the distance $d = 22.5$ mm ($B \sim 200$ mT) are provided for each strip.”... “The corresponding origami and magnetization pattern of each magneto-origami strip is shown below the experimental pictures. The quantitative control scheme is also provided.”

On Page 11, we add “We demonstrate such rolling locomotion by navigating the spring actuator through several obstructions including stairs, obstacles, narrow path, and groove by controlling the distance d , rotation angle α and position of the magnet in Figure 5f (Supplementary Movie 6).”

Revised Figure 4. Sequential folding and unfolding of the magneto-origami strips. (a-c) The schematic illustrations and experimental demonstrations of magneto-origami strips programmed with alternating magnetic polarity patterns. The magneto-origami strips can be sequentially folded into predefined 3D shapes: (a) lamellar, (b) triangular, and (c) square, by moving and rotating the permanent magnet. Quantitative control

schemes are provided in which rational angle α (left axis, blue curve) and distance d between the magnet and magneto-origami strip is plotted as a function of the magnet position. (d) Experimental demonstrations and quantitative control schemes of sequential folding of magneto-origami strips into letters “b”, “m”, and “e”.

Revised Figure 5. Actuations of magneto-origami spring actuator. (a) The spring actuator can contract when the magnet is placed in its axial direction. (b) The contraction ratio $(L_0 - L)/L_0$ of the spring actuator plotted as a function of magnetic field strength up to 220 mT. (c) The spring actuator remains 93 % of the original length L_0 after a 1000-cycle actuation ($\mathbf{B} \sim 220$ mT). (d) Bending behaviors of the spring actuator. Panels show bending angles are 0° , 45° , 90° , and 180° , respectively. (e) The 360° rotation behavior of a bent spring actuator by rotating the magnet surrounding the actuator. The yellow arrow represents the rotation direction of the rotating magnet. (f) Experimental demonstration of the rolling locomotion of the spring actuator by passing through several obstructions including stairs, obstacle, narrow path, and groove. The height of the stair, obstacle, and narrow path is 10 mm, 10 mm, and 12 mm, respectively. The width of the groove is 10 mm. Quantitative control schemes are provided in which rotation angle α (left axis, blue curve) and distance d between the magnet and magneto-origami spring actuator is plotted as a function of the magnet position.

Comment 4.

The phrase “2D-to-4D fabrication” in the title is confusing. In the literature, people use 4D fabrication/ printing to refer to a process of fabrication that involves the product being produced in a premature 3D form, and then changing shape over time to another 3D form in its service state. Hence the name means initial (3D shape + time) to new 3D shape. However, if 2D and 4D are put together, then they must refer to the same thing. Clearly, 2D refers to the geometry of the initial configuration of the origami machines (i.e. being flat), but 4D refers to a process, which causes a mismatch of subject and thus is inappropriate. The authors should call their approach either 4D fabrication, or 2D-to-3D fabrication, but not 2D-to-4D.

Response 4.

We thank the reviewer for pointing this out. We have eliminated the term “2D-to-4D” in the revised manuscript and changed the title to be “**High-throughput Fabrication of Soft Magneto-origami Machines**”

Comment 5.

How is the failure rate in these experiments? For example, failure of fabrication, failure of control, etc. It is important to provide a whole picture of the research to the readers, which is important for a scientific report.

Response 5.

Referring to **Response 1**, we have built the roll-to-roll platform and achieved an automated fabrication process of 2D magnet sheet patterns for desired magneto-origami machines. This roll-to-roll platform currently shows a 100% success rate of creating the 2D magnetic sheet patterns as showcased in the revised **Figure 1g**.

Referring to **Response 2**, statistical studies of making single-crease magneto-origami machine and multi-crease Miura magneto-origami have been conducted. All samples are successfully made and show small difference in both initial and deformed shapes in the magnetic fields up to 200 mT. Therefore, we conclude that failure rate in current work is negligibly small.

Comment 6.

Page 3, line 49: the elasto-plastic property of paper is very complex, it is not convincing to me that the authors could reliably “harness the predesigned plastic folding.”

Response 6.

We agree with the reviewer and have rephrased in the revised manuscript “**By harnessing the folding process, origami transforms a planar sheet into a 3D architecture**”.

Comment 7.

Page 3, line 52: please be more specific on what do you mean by “a large degree of recovery.”

Response 7.

To alleviate confusion, we have deleted such description in the revised manuscript.

Comment 8.

Page 4, line 78: please be more quantitative when describing “the folded origami recovers partially to the rest state.” What are the rest angles of the folding creases? How do you make sure that they can be repeatedly achieved in fabrication?

Response 8.

Two statistical studies have been conducted to show that the magneto-origami machines with different geometries can be consistently fabricated. Details can be found in our **Response 2**. For your convenience, we list key revised context here.

On Page 7, we add “ Note that the rest angle equals 130° (i.e., $\beta_0 = 130^\circ$) and when the magnetic field strength is 200 mT, the magneto-origami is completely folded (i.e., $\beta = 180^\circ$).”...“ To verify that such a manual folding process has little influence on the magneto-origami machines, two statistical studies of 8 volunteers each of whom folds 7 samples have been conducted. First, a single-crease magneto-origami machine is selected, and the average angle $\beta_0 = 130^\circ$ with a standard deviation of 4° at rest state are consistently achieved (Supplementary Figure 11a-b). Second, a multi-crease Miura magneto-origami is selected owing to its complex folding process (Figure 2d, Supplementary Figure 11 and Supplementary Movie 3). The freshly folded Miura magneto-origami have an average width of 2.4 mm with a very small standard deviation of 0.14 mm. Notably, such a folding process only takes about 140 s per sample, indicating high 3D fabrication efficiency of origami technique.”

Comment 9.

Page 6, line 124: the bistable snapping behavior of the square origami twist is an interesting intrinsic property arising from its geometry. It would be nice to see some simulations and analysis on how this property interacts with the magnetic forces applied onto the origami panels. Is the bistability enhanced by the magnetic forces? Or hindered but still preserved to some extent?

Response 9.

This square-twist origami was initially presented and discussed in literature (Silverberg, J., Na, JH., Evans, A. *et al.* Nature Materials, 14, 389–393 (2015)). The bistable behavior is an intrinsic property of the structure as shown by the stretch force vs. extension curve in Figure R1. When it is folded, one needs to apply a certain force to unfold it (red curve).

[REDACTED]

Figure R1. Tensile force vs. extension curve (adapted from Nature Materilas 14, 389–393 (2015))

In our work, we fabricated such square origami using the magnetic sheet and achieved magnetically deployable twisting. When it is folded, reversing the magnetic field direction will not reopen it. To answer your questions, we added experiments by measuring the stretch force vs. extension of four square-twist origami made by (1) pure paper; (2) nonmagnetized composite on paper; (3) magnetized composite on paper; (4) magetized composite on paper in field. Results are shown in Supplementary Figure 12. As consistent with Nature Materilas 14, 389–393 (2015)), all four sample show the bistable behavior. The nonmagnetized sample (red curve) shows similar peak force to the pure paper sample (blue curve). When the magnetic composite is magnetized, the peak force is getting higher due to the attractive interaction between layers in the folded state (green curve). In particular, when subjected to magentic field, the peak force is enhanced markedly because of the attractive force between magnetized sheet and the magnetic field. Therefore, magentic field enhances the bistable behavior of the magneto-origami square-twist.

On Page 9, we add “Due to its unique geometry³⁸, the magneto-origami square-twist exhibits a bistable behavior. At the folded state, it is stable and one needs to apply a certain force to unfold it (Supplementary Figure 12). The required force is further enhanced when the magneto-origami square-twist is in a magnetic field because the magnetic field will provide additional magnetic forces preventing the unfolding. When it is unfolded, it can remain flat in the absence of a magnetic field. Applying the actuation magnetic field will trigger the snap transition from unfolded state to folded state as manifested by the abrupt change in the diagonal length δ in Figure 3b.”

Supplementary Figure 12. Bistable behavior of magneto-origami square-twist (a) Schematic of

experimental setup to characterize the force (F)-extension (ΔL) relationship. (b) Tensile force as a function of normalized extension $\Delta L/L$ of magneto-origami square-twist made by (1) pure paper; (2) nonmagnetized composite on paper; (3) magnetized composite on paper; (4) magnetized composite on paper in field \mathbf{B} .

Reviewer # 2

General comment

The manuscript introduces a new method of manufacturing sheets of magnetically-responsive materials that can be cut with origami patterns and externally actuated to make prescribed motions. The process could be extended to a roll-to-roll manufacture of functional remote-controlled robots. This is an interesting idea, and the authors used their new process to implement some impressive improvements on the state of the art of magnetically-actuated robots. I have some comments that I think would improve the paper, presented as general comments here and specific comments in subsequent sections denoted with double-hash marks ("##").

Response

We sincerely thank the reviewer for recognizing the novelty of our work and your comments are addressed below in detail.

Comment 1.

1. For completeness, please compare more quantitatively and directly with the most relevant prior art.

1.1 Most importantly, the authors should discuss reference 9 in more detail. H. Song et al. present a reprogrammable magnetically-actuated origami sheet.

1.2 Additionally, to refresh myself on this niche area (magnetically-responsive origami robots), I did a few quick search on Google Scholar with naïve terms "magnet origami" and "magnetic origami"... and found several un-mentioned references that serve a similar purpose (small magnetically-actuated, possibly bio-compatible robots). While extensive introductions are not in the style of Nature Communications, I think - especially given the inclusion of Table S1 - more thorough description of prior art is warranted.

Daniella Rus has several papers in this area

[1] <https://ieeexplore.ieee.org/stamp/stamp.jsp?tp=&arnumber=7139386>

[2] <https://robotics.sciencemag.org/content/robotics/2/10/eaao4369.full.pdf>

[3] <https://ieeexplore.ieee.org/stamp/stamp.jsp?tp=&arnumber=7989560>

[4] <https://ieeexplore.ieee.org/stamp/stamp.jsp?tp=&arnumber=7487222>

Other authors

[5] <https://onlinelibrary.wiley.com/doi/epdf/10.1002/adfm.201904977>

[6] <https://pubs.rsc.org/en/content/articlehtml/2014/tc/c4tc00787e>

1.3 Finally, I think the comparison with other methods in Table S1 is over-simplified and un-justified, especially in light of its limited scope. Why did the authors omit reference 9 from this comparison? The current presentation makes it seem like the chosen competitors were hand-picked to create a favorable comparison, rather than dispassionately evaluate the current work in context of the state of the art.

(Specific comments:) - I recommend doing a more in-depth comparison to other magnetically-actuated origami robots, to clarify the contribution further.

Response 1.

We have significantly revised the Introduction section and thoroughly discussed the existing works as suggested. We also revised Supplementary Table 1 to give an in-depth comparison between our work and existing works.

Supplementary Table 1. Comparison between this work and existing soft magneto-active machines.

Fabrication Method	Materials	Magnetization encoding	Structural complexity	Time cost	Production scale	Locomotion modes	Reference
3D printing	Hard magnetic particle @ polymer matrix	Sequential encoding while printing	Low	Intermediate	Intermediate	Bending/rolling /walking	(13-15)
Laser printing	Hard magnetic particle @ polymer matrix	Sequential encoding while heating	Low	High	Small	Bending/rolling /walking	(16-17)
2D template molding	Soft magnetic particle @ polymer matrix	Encoding at one time while curing	Low	High	Intermediate	Bending/Crawling	(12)
Voxel assembly	NdFeB @ PDMS voxel +Dragon skin voxel +Ecoflex bonding agent	Encoding before assembly	High	High	Small	Bending/twisting /contraction	(18)
Transfer printing	NdFeB @ PDMS +Stamp film +Adhesive	Encoding before assembly	Intermediate	High	Small	Bending/crawling	(19)
Origami +Assembly	Permanent magnets +Origami (e.g, polystyrene)	Permanent magnet orientation	High	Low	Intermediate	Contraction/deploying /bending/twisting	(29-33)
Origami +Magnetic Spray	Fe @ PVA&Gluten + Paper	Sequential encoding while curing	High	High	Small	Folding/rolling /walking/crawling	(35)
Origami	Magnetic sheet: NeFeB+ PEG shell @ Ecoflex	Encoding at one time while heating	Intermediate	High	Small	Bending/folding	(34)
Origami + Roll-to-roll	Magnetic sheet: NeFeB @ Ecoflex on paper	Encoding at one time after curing	High	Low	Large	Bending/folding/ contraction/ deploying/rolling	This work

On Page 2, we add “Despite various strategies for fabricating soft magneto-active machines have been proposed (e.g., template molding,¹² extrusion-based 3D printing,¹³⁻¹⁵ laser printing,^{16, 17} voxel assembly¹⁸, and transfer printing¹⁹), existing fabrication methods suffer from limitations of the low structural complexity and/or time-consuming process (Supplementary Table 1). For example, the template molding method may only make 2D structures^{12, 20} and the extrusion-based 3D printing fails to print complex structures due to the viscosity and the die swell of the composite ink.²¹ The laser programming method usually requires a relatively long time to encode magnetic polarity patterns in a large area.^{16, 22} Therefore, the fast fabrication of soft magneto-active machines with customized architectures yet remains an unresolved challenge.”

On Page 3, we add “A simple way to empower origami with magnetic responsiveness is to directly adhere permanent magnets to the origami.²⁹ For example, Rus’s group has created a few magnet-based origami robots and demonstrated the capability of locomotion³⁰, metamorphosis,³¹ swimming,³² and drug delivery.³³ However, the presence of permanent magnets limits the softness and miniaturization of the robot, while increasing the undesired risk of magnet break-off in clinic applications.³⁴ Recently, Song *et al.* developed a soft magnetic sheet that can be folded into various magneto-origami structures³⁵. But without a backing layer, such a pure magnetic sheet exhibits an inferior structural resolution to its counterpart of paper origami, especially at folding creases.³⁵ Yang *et al.* directly sprayed uncured magnetic composite to the origami, yet still experienced prolonged time in sequentially curing/magnetizing each part of the origami.³⁶ So far, to the best of our knowledge, no fabrication method leverages structural complexity, softness, resolution, and fast production of magneto-origami machines at a large scale.

Here, we propose a facile fabrication strategy that can fast construct soft magneto-active machines at a large

scale by incorporating roll-to-roll processing of 2D patterns and 3D origami folding (Figure 1). By simply coating and curing a layer of magnetic composite (i.e., dispersing hard-magnetic microparticles in the polymer matrix) on a piece of raw paper, we create a paper-like soft magnetic sheet that can be directly folded into magneto-origami machines with customized geometries and high resolution. The magnetic polarity of the magneto-origami machine is quickly encoded at one time by a strong impulse magnetizing field ($H \sim 3T$). When subjected to an actuation magnetic field, the magneto-origami machine can achieve on-demand shape-morphing. We demonstrate a set of magneto-origami machines that can fold, bend, roll and walk with potential applications of deployment, object manipulation, and locomotion. Compared with existing soft magneto-active machines, our roll-to-roll platform allows for the automated fabrication of 2D magnetic patterns, followed by a quick magnetization process and direct folding, giving rise to facile and high-throughput production of magneto-active machines in various geometries and multimodal shape-morphing capabilities (Supplementary Table 1).”

Comment 2.

The claims (and figures) about roll-to-roll manufacturing are unjustified and/or misleading. Please show more details about the specific process used to make the robots, and remove all schematics showing theoretically-possible processing steps (primarily referring to Figure 1a), or at least make it clearer what was done vs. what could be done. This is especially important, since the manufacturing method is the main contribution of the manuscript. Make your contributions shine, and avoid over-claiming. If you do this well, readers will be able to more readily learn from your work and integrate the introduced concepts into their studies --- which is the main value in scientific publications.

Response 2.

To strengthen our work, we have built a roll-to-roll platform to achieve an automated fabrication of magnetic sheets with desired patterns as shown in the revised **Figure 1b-1e**.

On Page 4, we add “**Fabrication of soft magneto-origami machines.** The magnetic composite consists of magnetizable neodymium-iron-boron (NdFeB) particles (average diameter $\sim 30 \mu\text{m}$) embedded in the polymer resin (i.e., Ecoflex 00-10, Smooth-On) with a particle volume fraction of 25%. The 2D magnetic sheet is made by coating and curing a layer of magnetic composite on a piece of raw paper (Figure 1a, Supplementary Figure 1). To achieve an automated fabrication of 2D magnetic sheets, we build a roll-to-roll platform as shown in Figure 1b-1e. The major components of roll-to-roll platform consist of a roller of raw paper, a scraper (Figure 1c) that creates an even layer of uncured magnetic composite on the raw paper, a heater (Figure 1e) that cures the magnetic composite layer, a laser machine (Figure 1d) that cuts the magnetic sheet into designed patterns, and an end roller. After curing, the designed 2D pattern is cut by a laser machine with uncut connections to allow for an easy tear-off (Supplementary Figure 1, Supplementary Movie 1). The rolling velocity, the temperature of the heater, and the pattern of the magnetic sheet in the roll-to-roll process can be readily adjusted in an automated manner, which paves the way for the high-throughput fabrication of magneto-

origami machines. Supplementary Movie 1 shows the detailed roll-to-roll process in which the production rate is 100 mm/min.”

Revised Figure 1. Fabrication of magneto-origami machines. (a) Schematic illustration of fabrication processes of the patterned magnetic sheet by roll-to-roll method, folding/magnetizing, and partial recovery. (b)-(e) Experimental pictures of (b) laser cutting, (c) scraper, (d) pattern, (e) curing heater of the roll-to-roll platform to fabricate patterned magnetic sheets. (f) Images showing a roll of raw paper and magnetic sheet with a 2D origami pattern. (g) Magneto-origami machines with customized geometries and high structural resolution can be consistently fabricated.

Comment 3.

Several of the figures (such as 1a, 1d-e, 2 all, 3 all) are confusing, which detracts from the readability of the manuscript.

Response 3.

Figure 1 has been significantly revised to clarify the fabrication process. Please refer to our **Response 2** above for detailed modification.

The revised Figure 2 now is the theoretical discussion, numerical simulation, and reproducibility test of the magneto-origami machines.

Revised Figure 2. Theoretical analysis, finite element simulations, and reproducibility test of magneto-origami machines. (a) Schematic illustration showing the shape-morphing of the single-crease magneto-origami actuated by a magnetic field. Equilibrium is reached when the total driving magnetic torque T_m is equal to elastic torque T_e . (b) Comparison of deformation of single-crease origami between experiments and finite element simulations. (c) Rotation angle β as a function of applied magnetic field strength up to 200 mT. (d) Example of folding Miura magneto-origami from the 2D magnetic sheet, in which red solid lines represent the mountain folds while black dashed lines denote valley folds. The left insert shows the magnetic polarity pattern where cross means magnetization pointing inward while blue represents magnetization pointing outward. The right insert shows the finalized Miura magneto-origami. (e) Demonstration of shape morphing of Miura magneto-origami under different magnetic fields. (f) Statistical analysis of reproducibility and performance evaluation of Miura magneto-origami fabricated by 8 volunteers each of whom folded 7 samples.

On Page 7, we add “**Shape-morphing mechanism of magneto-origami machines.** Fully magnetized magneto-origami machines retain the residual magnetic polarity patterns, i.e., magnetization \mathbf{M} . When subjected to an actuation magnetic field, magneto-origami machines can deform and move. In this work, we select a cuboid permanent magnet (NdFeB, dimension: 50 mm × 25 mm × 20 mm, residual magnetization 103 kA/m) as the magnetic actuation source (Supplementary Figure 8) because it is easier to handle and can generate stronger magnetic fields (up to 200 mT in this work) than electromagnetic coils. By varying the position/orientation of the cuboid magnet, the magnetic field strength \mathbf{B} around the magneto-origami machine can be effectively changed (Supplementary Figure 8). For example, the magnetic field strength on the centerline of the cuboid magnet can reach up to 200 mT when the distance between the center of the magnet and the magneto-origami machine is around 22.5 mm (Supplementary Figure 8)

To elucidate the magnetically controllable shape-morphing mechanism, here we adopt a single-crease magneto-origami machine as a representative example (Figure 2a). Upon actuation, the upper panel of the magneto-origami machine rotates around the crease due to the driving force, i.e., the magnetic torque density $\mathbf{M} \times \mathbf{B}$ and body force $(\nabla \mathbf{B})\mathbf{M}$ where operator “ \times ” and “ ∇ ” represents the cross-product and gradient, respectively. Therefore, the total magnetic torque that deforms the upper panel can be calculated as

$$\mathbf{T}_m = \int_{\Omega} [\mathbf{M} \times \mathbf{B} + \mathbf{r} \times (\nabla \mathbf{B})\mathbf{M}] dv \quad (1)$$

where Ω represents the 3D domain of the upper panel and \mathbf{r} is the position vector of a spatial point on the upper panel with respect to the crease (Figure 2a). According to the literature² and our experimental validation (Supplementary Figure 6), the folding crease behaves like an elastic hinge in which the total elastic torque is linearly proportional to angle change of the crease, i.e.,

$$\mathbf{T}_e = kL(\beta - \beta_0) \quad (2)$$

where $k = 1.47 \mu\text{N/deg}$ represents the experimentally measured torsional constant of the crease per unit width, L , $\beta_0 \approx 130^\circ$ and β denote the width, the rest angle, and folding angle of single-crease magneto-origami, respectively (Figure 2a, Supplementary Figure 6). Therefore, the deformed shape of the magneto-origami machine can be found when the equilibrium is reached, i.e.,

$$\mathbf{T}_m = \mathbf{T}_e \quad (3)$$

Considering that magnetic fields around a cuboid magnet are nonuniform, Eq. (3) is solved by finite element analysis (FEA) (Supplementary Figure 9). The predicted folding angle β by FEA under various magnetic fields up to 200 mT are validated by experimental measurements (Figure 2b, 2c). Note that when the magnetic field strength is 200 mT, the magneto-origami is completely folded. Upon removal of the actuation field, the magneto-origami machine rapidly recovers the rest state. This reversible folding and unfolding process is shown in Movie 2.”

Reproductivity tests of magneto-origami machines together with corresponding descriptions can be found in our **Response 3** below.

Figure 3 has been revised with further descriptions. We also added Supplementary Figure 12 to further explain the bistable behavior of the magneto-origami square-twist.

Revised Figure 3. Folding/unfolding of deployable magneto-origami machines under a magnetic field of 200 mT. (a) The unfolding process of the magneto-origami starshade that is magnetized in the flat state. The red marker indicates the movement of a marginal point during the deployment process. The blue sign indicates a magnetization pointing outward. (b) The diagonal length δ of the magneto-origami starshade during the deployment process. (c) Twisting and folding process of the magneto-origami square-twist that is magnetized at the folded state. (d) Diagonal length δ of magneto-origami square-twist during the folding process. Magneto-origami machines in both (a) and (b) have a bistable behavior with a snap transition between folded and unfolded states.

Supplementary Figure 12. Bistable behavior of magneto-origami square-twist (a) Schematic of the experimental setup to characterize the force (F)-extension (ΔL) relationship. (b) Tensile force as a function of normalized extension $\Delta L/L$ of magneto-origami square-twist made by (1) pure paper; (2) nonmagnetized composite on paper; (3) magnetized composite on paper; (4) magnetized composite on paper in field B .

On Page 9, we add “Here we present two deployable magneto-origami machines that can morph to different configurations by applying a magnetic field ($B \sim 200$ mT) in Figure 3 and Supplementary Movie 4. First, we present a magneto-origami square-twist in Figure 3a. The magneto-origami square-twist consists of alternating square and rhombus facets with tailored mountain and valley folds. Being magnetized at the folded state, it

has a fourfold rotational symmetry in the magnetic polarity patterns. Due to its unique geometry³⁸, the magneto-origami square-twist exhibits a bistable behavior. At the folded state, it is stable and one needs to apply a certain force to unfold it (Supplementary Figure 12). The required force is further enhanced when the magneto-origami square-twist is in a magnetic field because the magnetic field will provide additional magnetic forces preventing the unfolding. When it is unfolded, it can remain flat in the absence of a magnetic field. Applying the actuation magnetic field will trigger the snap transition from unfolded state to folded state as manifested by the abrupt change in the diagonal length δ in Figure 3b. Next, a deployable magneto-origami starshade is developed in reminiscence of the starshade that blocks the glare of stars for the space telescopes (Figure 3c). Different from the square-twist that is magnetized in its folded state, the magnetic sheet is first magnetized and then folded into a starshade (Further discussions on this fabrication method are provided in DISCUSSION). Under the magnetic actuation, the magneto-origami starshade gradually expands while rotating around its center (shown by the red marker at the margin in Figure 3c). After 7.5 s, the magneto-origami starshade quickly unfurls to a nearly flat sheet evidenced by the steep increase of its diagonal length δ (Figure 3d). It is worth noting that both the folding of the magneto-origami square-twist and the unfolding of the magneto-origami starshades complete in a short time of 8 s with a change in the area by two folds. The large area change is intrinsically enabled by the origami structure and the on-demand shape-morphing is highly desired for various applications such as energy harvesting^{39, 40}.

We added a quantitative control scheme in the revised Figure 4, 5 to help the reader understand the control of the magneto-origami strips.

On Page 9, we add “Corresponding quantitative control scheme by adjusting the position and rotation angle α of the magnet (see insert in Figure 3a-3c, Supplementary Figure 8c) while keeping the distance $d = 22.5$ mm ($B \sim 200$ mT) are provided for each strip.”... “The corresponding origami and magnetization pattern of each magneto-origami strip is shown below the experimental pictures. The quantitative control scheme is also provided.”

On Page 11, we add “We demonstrate such rolling locomotion by navigating the spring actuator through several obstructions including stairs, obstacles, narrow path, and groove by controlling the distance d , rotation angle α and position of the magnet in Figure 5f (Supplementary Movie 6).”

Revised Figure 4. Sequential folding and unfolding of the magneto-origami strips. (a-c) The schematic illustrations and experimental demonstrations of magneto-origami strips programmed with alternating magnetic polarity patterns. The magneto-origami strips can be sequentially folded into predefined 3D shapes: (a) lamellar, (b) triangular, and (c) square, by moving and rotating the permanent magnet. Quantitative control schemes are provided in which rotation angle α (left axis, blue curve) and distance d between the magnet and magneto-origami strip is plotted as a function of the magnet position. (d) Experimental demonstrations and quantitative control schemes of sequential folding of magneto-origami strips into letters “b”, “m”, and “e”.

Revised Figure 5. Actuations of the magneto-origami spring actuator. (a) The spring actuator can contract when the magnet is placed in its axial direction. (b) The contraction ratio $(L_0 - L)/L_0$ of the spring actuator plotted as a function of magnetic field strength up to 220 mT. (c) The spring actuator remains 93% of the original length L_0 after a 1000-cycle actuation ($B \sim 220$ mT). (d) Bending behaviors of the spring actuator. Panels show bending angles are 0° , 45° , 90° , 180° , respectively. (e) The 360° rotation behavior of a bent spring actuator by rotating the B magnet surrounding the actuator. The yellow arrow represents the rotation direction of the rotating magnet. (f) Experimental demonstration of the rolling locomotion of the spring actuator by passing through several obstructions including stairs, obstacles, narrow path, and groove. The height of the stair, obstacle, and narrow path is 10 mm, 10 mm, and 12 mm, respectively. The width of the groove is 10 mm. Quantitative control schemes are provided in which rotation angle α (left axis, blue curve) and distance d between the magnet and magneto-origami strip is plotted as a function of the magnet position.

Comment 4.

I would suggest doing multiple trials with multiple samples for most of the experiments, so confidence intervals may be obtained and readers can gain confidence in the repeatability of the manufacturing process. In absence of explicit discussion of repeatability problems, whenever one sample is shown, some readers will wonder whether the authors simply chose to present the only sample that worked.

Response 4.

Thank you for your suggestion. To show that the manual folding process has little influence on the magneto-origami machines, statistical studies with 8 volunteers each of whom folds 7 samples have been conducted (revised Figure 2d-2f). Here, we choose a single-crease magneto origami and multi-crease Miura magneto-origami as representative examples. Statistics in Supplementary Figure 10 show that both the rest angle of the single-crease magneto-origami and width of Miura magneto-origami at rest state are almost the same among all volunteers. In addition, we also evaluate the shape-morphing performance of the Miura magneto-origami when actuated by a magnetic field. Results in revised Figure 2f clearly indicate a small variation between different samples.

Revised Figure 2. (d) Example of folding Miura magneto-origami from the 2D magnetic sheet in which red solid lines represent the mountain folds while black dashed lines denote valley folds. The left insert shows the magnetic polarity pattern where cross means magnetization pointing inward while blue represents magnetization pointing outward. The right insert shows the finalized Miura magneto-origami. (e) Demonstration of shape morphing of Miura magneto-origami under different magnetic fields. (f) Statistical analysis of reproductivity and performance evaluation of Miura magneto-origami fabricated by 8 volunteers each of whom folded 7 samples.

On Page 7, we add “To verify that such a manual folding process has little influence on the magneto-origami machines, two statistical studies of 8 volunteers each of whom folds 7 samples have been conducted. First, a single-crease magneto-origami machine is selected, and the average angle $\beta_0 = 130^\circ$ with a standard deviation of 4° at rest state are consistently achieved (Supplementary Figure 11a-b). Second, a multi-crease Miura magneto-origami is selected owing to its complex folding process (Figure 2d, Supplementary Figure 11 and Supplementary Movie 3). The freshly folded Miura magneto-origami have an average width of 2.4 mm with a very small standard deviation of 0.14 mm. Notably, such a folding process only takes about 140 s per sample, indicating high 3D fabrication efficiency of origami technique. In addition, when actuated in a

magnetic field up to 200 mT (Figure 2e), they exhibit small differences in the deformed width (Figure 2f) and excellent reversibility when the magnetic field is removed (Supplementary Movie 3). These statistical studies manifest that the proposed fabrication method by roll-to-roll process and manual folding can construct magneto-origami machines with consistent geometries that enable controllable shape-morphing under magnetic actuation”

Supplementary Figure 11. Statistical studies of magneto-origami machines. (a) Optical image of a single-crease magneto-origami at the rest state. (b) The average and standard deviation of the rest angle of the single-crease magneto-origami is 130° and 4° , respectively. (c) Optical image of a Miura magneto-origami machine. (b) The average and standard deviation of the length of all the samples is approximately 2.4 mm and 0.14 mm, respectively. Such a folding process only takes about 140 s.

Comment 5.

The materials and methods section was lacking some important details. I commented on a few below, but I encourage the authors to attempt to repeat their experiments solely based on the materials and methods. Alternatively, ask a colleague to draw the whole process on a whiteboard based on the M&M section. This will highlight even more deficiencies than I showed.

Response 5.

We thank the reviewer for pointing out this issue. We have carefully revised the Materials and Methods section.

Preparation of magnetic composite. The Ecoflex 00-10 was purchased from Smooth-On, Inc. NeFeB microparticles (average diameter: $30 \mu\text{m}$) was obtained from Magnequench Co. Ltd. The composite ink was firstly prepared by mixing Ecoflex 00-10 with NdFeB microparticles with a particle volume fraction of 25%, followed by sufficient stirring and degassing.

Roll-to-roll fabrication of magnetic sheets. The roll-to-roll scraping process is realized by a scraping coating machine (MSK-AFA-EC200, Shenzhen Kejing, China). The roll paper with the magnetic composite coating is scraped evenly with a scraper, and the distance between the scraper and the paper surface is set to 250 microns. The paper is heated by the dryer, the effective heating length of the dryer is 0.5 m, the temperature is set to 75 °C, the delivery speed of the paper is 100 mm/min, and the paint is dried after this step. The roll paper continues to go through the laser cutting machine (PWJDD-20W, Pinzan, China) for cutting according to the pre-designed pattern. After passing through each transfer roller, in turn, the cut roll is finally rolled on the collection roller.

Magneto-origami machines. The fabrication of the magneto-origami machines followed the same tearing, folding and magnetizing process. The patterns were designed using SolidWorks (Version 2016, Dassault Systems, France) and engraved via a laser cutter. The pattern was manually folded into 3D shape and subsequently magnetized by an impulse magnetic field (3T) using a magnetizer (MA-2030, Shenzhen JiuJu Company, China). For the mechanical 8-3 encoder, to make the spring actuator conductive, the magnetic paper was coated with an Au film by radiofrequency magnetron sputtering deposition (VTC300, Shenyang Micro Technology Co., Ltd, China). **Then the conductive spring can be utilized as the connection in the logic system.** The ME-robot was fabricated by transferring a layer of copper wire on the magnetic sheet. The copper wire was obtained by laser cutting a copper layer (thickness ~50 μm) and removing the needless part. The electronic components (rectifier, capacitance, voltage stabilizer) and the copper wire were joined by soldering.

Comment 6.

Introduction

- "mass and fast fabrication" -- this "mass" sounds like "kilograms" rather than "high-throughput" or "large-quantity"

- What makes these 4D? I have this criticism of most other studies' claims of 4D as well, but for consistency I must bring it up. This seems like 2D-3D deployment. Please justify the choice of this 4D term.

Response 6.

To alleviate concerns, we have deleted the terms “mass production”, “2D-to-4D” in the revised manuscript. In particular, we have revised the title of our work as “**High-throughput Fabrication of Soft Magneto-origami machines.**”

Comment 7.

Results and Discussion

-(1) "self-folding" is inapplicable here. The films are actuated externally. In order to fold the structures multiple times, you need an external 3-DOF system to {lift, move, and rotate} the external magnet.

-(2) "magneto-origami electronic robot" - What does that mean? What makes this robot electronic, in contrast with other magnetically-actuated sheets (such as those referenced as refs. 1, 3, 9, 13, 14)?

Response 7.

- (1) We have revised the term “self-folding” to be “folding” in the revised manuscript.
- (2) There is a wireless charging device circuit on the sheet. When actuated by an alternating magnetic field from an electromagnetic coil, the printed spiral wire can light up the LED wirelessly through electromagnetic induction. To alleviate concerns, we have revised it to be “magneto-origami charging robot” in the revised manuscript.

Comment 8.

Materials and Methods

- (1) "Finally, a bilayer magnetic paper was fabricated." -- how?
- (2) Why was the spring coated with Au? Is this for making connections in the logic device? I'd recommend explicitly stating this
- (3) How were the magnets moved for the various examples shown in the paper? By hand? With a robot arm? With a multi-DOF gantry/stage?

Response 8.

(1) We have revised the **Materials and Methods**. Please refer to our **Response 5** for details.

(2) Yes. Coating the spring with Au is to make it electrically conductive.

On Page 12, we add “To endow the conductivity of the encoder, the spring actuator is coated with a thin layer of gold.”

(3) Magnets are moved by hand with care. The magnetic field strength is simultaneously measured by a Tesla meter (CH-Hall Model 1500, Magnetic Technology, China). We have added Revised Figure 2 (see **Response 3**) and Supplementary Figure 6 to show the control parameters of the magnet (the distance d and the rotation angle α). We also provide the quantitative control scheme in Figure 4 and Figure 5 (see **Response 3**).

To alleviate your concerns, we also add some discussions on the limitation of our current work and provided some future perspectives in DISCUSSION.

On Page 6, we add “In this work, we select a cuboid permanent magnet (NdFeB, dimension: 50 mm × 25 mm × 20 mm, residual magnetization 103 kA/m) as the magnetic actuation source (Supplementary Figure 8a) because it is easy to handle and can generate stronger magnetic fields (up to 200 mT in this work) than electromagnetic coils. By varying the position/orientation of the cuboid magnet, the magnetic field strength \$B\$ around the magneto-origami machine can be effectively changed (Supplementary Figure 8c). For example, the magnetic field strength on the centerline of the cuboid magnet can reach up to 200 mT when the distance between the center of the magnet and the magneto-origami machine is around 22.5 mm (Supplementary Figure 8d).”

On Page 13, we add “In addition, actuating magneto-origami machines by the cuboid magnet is performed by human hands. More accurate control can be done by integrating the cuboid magnet with a robotic arm.”

Supplementary Figure 8. A cuboid permanent magnet for actuating magneto-origami machines. (a) Schematic of the cuboid magnet with 25 mm in width, 50 mm in length, and 25 mm in height. (b) The magnetic field around the cuboid magnet. The distribution and magnetization of the permanent magnet was obtained using COMSOL Multiphysics software under the same conditions including dimensions and magnetization. (c) Schematic of two parameters (distance d and angle α) to quantitatively control the permanent magnet. (d) The magnetic field strength as a function of distance d along the centerline of the magnet ($\alpha = 0^\circ$).

Comment 9.

Supplementary materials (mostly small comments)

- (1) How was the magnetic field in Figure S5 measured?
- (2) Fig S7 - b x-axis should be "cycles". It might be helpful to calculate the mean and standard deviation of the peak and valley (max and min) angles.
- (3) Fig S8 is confusing. Can you mark the start, and explain what the squares are? Also, why did it go from 5x5 L-shape, down to 3x3 L-shape? Why not go to a single 1-square shape for maximum springiness? In the videos and other figures, it seems you went to a 1-square shape.
- (4) Figure S9: I think "voltage regulator" is a more readily-used term than "voltage stabilizer". Also, what's the point of the wireless charging (also relates to Figure 5b), if there are not onboard batteries? If the point of this is to show that circuits can be embedded onto the ME-robot, then I think vibrating in a pig stomach phantom and activating a light is a little strange of a demonstration. Something more natural for robotics might be adding a sensor suite, or a microcontroller plus some actuators and/or sensors. This is not a technical criticism, just a thought on the presentation / appeal to wider audiences.

Response 9.

- (1) The magnetic field strength is measured by a Tesla meter (CH-Hall Model 1500, Magnetic Technology, China).

(2) We have revised the figure and reported the mean and standard deviation.

Supplementary Figure 7. A single-fold magneto-origami machine under cyclic magnetic actuation. (a) Images of the single-fold magneto-origami machine under different cycles. (b) The angle of the single-fold magneto-origami machine almost remains almost unchanged after 1000 cycles. The mean and standard deviation of the maximum angle are 180° and 0° (fully folded), respectively. The mean and standard deviation of the min angle are 131° and 1° , respectively.

(3) We have revised the figure (now Supplementary Figure 13).

Supplementary Figure 13. Schematic illustration of the fabrication of magneto-origami spring actuator. First, the end sections of two magneto strips are glued together. Next, two magneto strips are alternatively folded. By repeating this, a magneto-origami spring actuator is eventually finished. The spring actuator was compressed tightly and magnetized by a strong pulsed magnetic field. When the compression was removed, the spring actuator returned to its rest state.

(4) We have changed the term "voltage stabilizer" to "voltage regulator" as suggested. Inspired by references

[Sun, Q., *et al.* A miniature robotic turtle with target tracking and wireless charging systems based on IPMCs. *IEEE Access*, 8, 187156-187164; Lu, H. *et al.*,. Battery-Less Soft Millirobot That Can Move, Sense, and Communicate Remotely by Coupling the Magnetic and Piezoelectric Effects. *Advanced Science*, 7(13), 2000069], we show our magneto-origami machine also wireless charging capability that can be further used as the energy source for the long-term application of implantable electronic devices (e.g., sensing and transmitting data). Here, LEDs are simplified representative electronic devices to show the charging performance and they can be readily replaced by other functional devices. Therefore, we believe that our magneto-origami machine can be further equipped with electronic-wise functionality if needed in the further.

To alleviate concerns, on Page 12, we add “Note that LEDs are simplified representative models to demonstrate the wireless charging capability. They can be readily replaced by other functional electronics. This demonstration clearly shows that utilizing the 3D-to-2D shape-morphing, the magneto-origami machine can have on-demand large area change. By incorporating an electronic circuit, it is further empowered with wireless charging capability that can potentially provide for the long-term application of implantable electronic devices.”

Comment 10.

- (1) (minor) The title slide has tons of words and is super difficult to read in the time given. I would suggest removing the affiliations, shrinking the author list, simplifying the background, moving the movie titles (such as "Movie S4: Sequential folding and unfolding of the magneto-strips") up to the top and larger.
- (2) (minor) Transitions between video segments are rather abrupt. Can you put fades, or transition slides?
- (3) (major) Video 8 - Does the robot even need a 4-legged configuration? Try this demonstration with a single larger spring to support the gripper. It appears to just be bouncing and moving to stay above the magnet (kind of like the kids toys that you drag a magnet under a steel ball to solve a maze)

Response 10.

- (1) We have revised the video as suggested.
- (2) We have put smooth transitions between segments.
- (3) The 4-legged configuration is needed. We have tested a single-legged configuration. We found that the robot could move driven by the magnetic attraction, but it could not unload the ball stably. Results are provided in Supplementary Figure 17 and Movie 9.

We add “In comparison, a single-legged magneto-origami robot was also tested. It can carry the ball and move driven the magnetic attraction. But it falls when unloading the ball at the destination.”

Supplementary Figure 17. Test of a single-legged robot. (a) A single-legged robot can carry the ball and move by magnetic attraction. (b) It falls when unloading the ball at the destination.

Reviewer # 3

General comment

The paper presents several interesting demonstrations of magneto responsive small robots based on different origami designs. However, compare to the recent advances in the relevant researches, novelty and contribution in the current manuscript are not convincing.

Response.

We sincerely thank the reviewer for carefully evaluating our work. We have significantly revised our manuscript and detailed responses/modifications are list below.

Comment 1.

Although the wording is different, there exist similar mechanisms on soft material by utilizing magnetic field-driven small robots capable of locomotion and shape morphing, e.g. paper-based, polymer-based, and so on. In there, the authors can find easy and convenient methods of attaching magneto-responsive polymer to a paper or magneto-responsive polymer itself to achieve 4D (in author's wording) fabrication. Moreover, these researches used both B-field and gradient fields altogether to perform the required motion based on a simple origami robot configuration. Also, the origami designs in this study are simple and some are already available in the literature. On these aspects, the novelty and contribution are weak, and might not be advanced compare to the state-of-the-art technology.

Response 1.

In this work, we empower the traditional paper origami with magnetically responsive functionality by simply replacing the raw paper with a paper-based magnetic sheet (revised **Figure 1a**). To strengthen our work, we have built a roll-to-roll platform to achieve an automated fabrication of magnetic sheets with desired patterns as shown in the revised **Figure 1b-1e**. The principle of designing magneto-origami patterns remains the same as that of traditional paper origami, which allow us to fabricate a broad range of 3Dmagneto-origami machines with high structural resolution and multimodal shape-morphing/locomotion modes as showcased in the revised **Figure 1g**.

In the revised manuscript, we have added Supplementary Table 1 and significantly modified **Introduction** to clearly describe the novelty of our work compared with state-of-the-art works and also provided some discussions in **DISCUSSION**. In summary, (1) from the material perspective, we are the first to prepare the magnetic sheet by coating and curing a layer of magnetic composite, which allow us to directly create 3D magneto-origami machines with a high structural resolution by folding; (2) from the fabrication perspective, we are the first to build the roll-to-roll platform, which paves the way for high-throughput fabrication; (3) from the functionality perspective, we demonstrate on-demand shape-morphing/locomotion of soft magneto-active machines with origami-enabled advantages such as large deformation (stretch ratio, bending angle, etc) and high area changes (folding to unfolding and vice verse).

Introduction

On Page 2, we add “**Despite various strategies for fabricating soft magneto-active machines have been**

proposed (e.g., template molding,¹² extrusion-based 3D printing,¹³⁻¹⁵ laser printing,^{16, 17} voxel assembly¹⁸, and transfer printing¹⁹), existing fabrication methods suffer from limitations of the low structural complexity and/or time-consuming process (Supplementary Table 1). For example, the template molding method may only make 2D structures^{12, 20} and the extrusion-based 3D printing fails to print complex structures due to the viscosity and the die swell of the composite ink.²¹ The laser programming method usually requires a relatively long time to encode magnetic polarity patterns in a large area.^{16, 22} Therefore, the fast fabrication of soft magneto-active machines with customized architectures yet remains an unresolved challenge.”

Supplementary Table 1. Comparison between this work and existing soft magneto-active machines.

Fabrication Method	Materials	Magnetization encoding	Structural complexity	Time cost	Production scale	Locomotion modes	Reference
3D printing	Hard magnetic particle @ polymer matrix	Sequential encoding while printing	Low	Intermediate	Intermediate	Bending/rolling /walking	(13-15)
Laser printing	Hard magnetic particle @ polymer matrix	Sequential encoding while heating	Low	High	Small	Bending/rolling /walking	(16-17)
2D template molding	Soft magnetic particle @ polymer matrix	Encoding at one time while curing	Low	High	Intermediate	Bending/Crawling	(12)
Voxel assembly	NdFeB @ PDMS voxel +Dragon skin voxel +Ecoflex bonding agent	Encoding before assembly	High	High	Small	Bending/twisting /contraction	(18)
Transfer printing	NdFeB @ PDMS +Stamp film +Adhesive	Encoding before assembly	Intermediate	High	Small	Bending/crawling	(19)
Origami +Assembly	Permanent magnets +Origami (e.g, polystyrene)	Permanent magnet orientation	High	Low	Intermediate	Contraction/deploying /bending/twisting	(29-33)
Origami +Magnetic Spray	Fe @ PVA&Gluten + Paper	Sequential encoding while curing	High	High	Small	Folding/rolling /walking/crawling	(35)
Origami	Magnetic sheet: NeFeB+ PEG shell @ Ecoflex	Encoding at one time while heating	Intermediate	High	Small	Bending/folding	(34)
Origami + Roll-to-roll	Magnetic sheet: NeFeB @ Ecoflex on paper	Encoding at one time after curing	High	Low	Large	Bending/folding/ contraction/ deploying/rolling	This work

On Page 3, we add “A simple way to empower origami with magnetic responsiveness is to directly adhere permanent magnets to the origami.²⁹ For example, Rus’s group has created a few magnet-based origami robots and demonstrated the capability of locomotion³⁰, metamorphosis,³¹ swimming,³² and drug delivery.³³ However, the presence of permanent magnets limits the softness and miniaturization of the robot, while increasing the undesired risk of magnet break-off in clinic applications.³⁴ Recently, Song *et al.* developed a soft magnetic sheet that can be folded into various magneto-origami structures³⁵. But without a backing layer, such a pure magnetic sheet exhibits an inferior structural resolution to its counterpart of paper origami, especially at folding creases.³⁵ Yang *et al.* directly sprayed uncured magnetic composite to the origami, yet still experienced prolonged time in sequentially curing/magnetizing each part of the origami.³⁶ So far, to the best of our knowledge, no fabrication method leverages structural complexity, softness, resolution, and fast production of magneto-origami machines at a large scale.

Here, we propose a facile fabrication strategy that can fast construct soft magneto-active machines at a large

scale by incorporating roll-to-roll processing of 2D patterns and 3D origami folding (Figure 1). By simply coating and curing a layer of magnetic composite (i.e., dispersing hard-magnetic microparticles in the polymer matrix) on a piece of raw paper, we create a paper-like soft magnetic sheet that can be directly folded into magneto-origami machines with customized geometries and high resolution. The magnetic polarity of the magneto-origami machine is quickly encoded at one time by a strong impulse magnetizing field ($H \sim 3T$). When subjected to an actuation magnetic field, the magneto-origami machine can achieve on-demand shape-morphing. We demonstrate a set of magneto-origami machines that can fold, bend, roll and walk with potential applications of deployment, object manipulation, and locomotion. Compared with existing soft magneto-active machines, our roll-to-roll platform allows for the automated fabrication of 2D magnetic patterns, followed by a quick magnetization process and direct folding, giving rise to facile and high-throughput production of magneto-active machines in various geometries and multimodal shape-morphing capabilities (Supplementary Table 1).”

Results

On Page 4, we add “**Fabrication of soft magneto-origami machines.** The magnetic composite consists of magnetizable neodymium-iron-boron (NdFeB) particles (average diameter $\sim 30 \mu\text{m}$) embedded in the polymer resin (i.e., Ecoflex 00-10, Smooth-On) with a particle volume fraction of 25%. The 2D magnetic sheet is made by coating and curing a layer of magnetic composite on a piece of raw paper (Figure 1a, Supplementary Figure 1). To achieve an automated fabrication of 2D magnetic sheets, we build a roll-to-roll platform as shown in Figure 1b-1e. The major components of roll-to-roll platform consist of a roller of raw paper, a scraper (Figure 1c) that creates an even layer of uncured magnetic composite on the raw paper, a heater (Figure 1e) that cures the magnetic composite layer, a laser machine (Figure 1d) that cuts the magnetic sheet into designed patterns, and an end roller. After curing, the designed 2D pattern is cut by a laser machine with uncut connections to allow for an easy tear-off (Supplementary Figure 1, Supplementary Movie 1). The rolling velocity, the temperature of the heater, and the pattern of the magnetic sheet in the roll-to-roll process can be readily adjusted in an automated manner, which paves the way for the high-throughput fabrication of magneto-origami machines. Supplementary Movie 1 shows the detailed roll-to-roll process in which the production rate is 100 mm/min.”

On Page 5, we add “To maintain the desired shape of the magneto-origami machine (e.g., to avoid shape collapse or excessive recovery), the mechanical properties of the magnetic sheet are carefully designed. Different recipes of magnetic sheets are tested by coating magnetic composite on aluminum (Al) foil, and polyimide (PI) membrane, and raw paper (Supplementary Figure 3). It is found that the Al-based magnetic sheet is too floppy to sustain the shape well (similar to the backing-free magnetic sheet developed by Song et al.³⁵), while the PI-based magnetic sheet is too stiff to be folded. By coating a $100 \mu\text{m}$ -thick magnetic composite layer on top of a $90 \mu\text{m}$ -thick raw paper (Figure 1f, Supplementary Figure 4), the paper-based magnetic sheet (Young’s modulus $\sim 7 \text{MPa}$, ultimate tensile stress $\sim 14 \text{MPa}$, Supplementary Figure 5) preserves the excellent folding capability of paper, allowing for consistent fabrication of magneto-origami machines in a broad spectrum of geometries with a high structural resolution, as showcased in Figure 1g.”

Revised Figure 1. Fabrication of magneto-origami machines. (a) Schematic illustration of fabrication processes of the patterned magnetic sheet by roll-to-roll method, folding/magnetizing, and partial recovery. (b)-(e) Experimental pictures of (b) laser cutting, (c) scraper, (d) pattern, (e) curing heater of the roll-to-roll platform to fabricate patterned magnetic sheets. (f) Images showing a roll of raw paper and magnetic sheet with a 2D origami pattern. (g) Various magneto-origami machines with customized geometries and high structural resolution can be consistently fabricated.

Supplementary Figure 3. Influence of different materials on fabrication and deformation of magneto-origami machines. (a) Magneto-origami machines using different materials, e.g., aluminum (Al) film, paper, and polyimide (PI) film. Folding performance of magneto-origami machines fabricated through paper (b) and PI (d), under a 200 mT magnetic field actuation.

Supplementary Figure 5. Mechanical properties of the magnetic sheet. Young's modulus is 7 MPa and ultimate tensile stress is 14 MPa.

DISCUSSION

On Page 13, we add “Origami, originating from the ancient Japanese art of paper, has nowadays been extensively explored as a fabrication strategy of engineering structures via a sequence of spatially organized folds. The key benefit of origami fabrication is the simple design principle and efficient creation of complex structures with customized geometries by constructing patterns in a plane and folding them into their final shape.²⁵ When utilized to design robots, origami robots exhibit intrinsic shape-morphing capability between 2D and 3D configurations. Enabled by multi-folds, the shape-morphing can have a large degree of deformation (e.g., stretching, contraction, bending), adapting the origami robots to different tasks and environments. In this work, we transform the conventional paper into a magnetic sheet by coating a thin layer of magnetic composite on the paper. The magnetic sheet preserves the paper-like foldability while imparting magnetic responsiveness to the magneto-origami machines it creates. We have built a roll-to-roll platform that achieves an automated fabrication of 2D magnetic patterns and demonstrate several 3D magneto-origami machines that can change their morphologies and execute locomotion tasks on demand, e.g., deploying, locking sequential folding, and cargo transportation.

Currently, magneto-origami machines are still in their infancy. Although substantial efforts have been made on designing folding patterns in literature, fabrication is still heavily associated with manual folding. In this regard, we have done two reproducibility tests to evaluate the finished shape and the shape-morphing behavior of the magneto-origami machine. Future work can be focused on how to develop the automated line for folding origami with consistent shape and performance. In addition, actuating magneto-origami machines by the cuboid magnet is performed by human hands. More accurate control can be done by integrating the cuboid magnet with a robotic arm. Furthermore, the magnetization of magneto-origami machines can also be encoded before they are folded, giving rise to different magnetic polarity patterns and shape-morphing behavior of the machine (Supplementary Figure 16, Supplementary Movie 8).”

The cuboid magnet is selected as the magnetic source to control the magneto-origami machines because (1) compared with electromagnets, the magnetic field strength of the cuboid magnet can be larger (up to 200 mT in our work); (2) handling cuboid magnet is simple and has been widely used in literature. We provided detailed **Response 2** below to discuss the theoretical and numerical analyses.

Comment 2.

The mechanism of magnet actuation is not depicted both in the experimental setup and supplementary video clip. Since the magnetic field intensity and gradient change with respect to the distance and magnetization direction, a detailed explanation of how the authors control the external magnetic field precisely (even with frequency) should be included. Moreover, the proposed mechanism should consider the effect of gradient field generated by a magnet for proper validation.

Response 2.

To elucidate the shape-morphing mechanism of the magneto-origami machines, we provide some theoretical discussions and numerical analysis in the revised manuscript by considering both the field and field gradient. We updated **Figure 2** and added **Supplementary Figure 6-7** in the revised manuscript.

Although currently the magnet is controlled manually, we believe it can be readily integrated to any digital control platform such as a robotic arm. To alleviate concerns, we discussed our limitations and future perspectives in the revised manuscript.

Revised Figure 2. Theoretical analysis, finite element simulations, and reproducibility test of magneto-origami machines. (a) Schematic illustration showing the shape-morphing of the single-crease magneto-origami actuated by a magnetic field. Equilibrium is reached when the total driving magnetic torque T_m is equal to elastic torque T_e . (b) Comparison of deformation of single-crease origami between experiments and finite element simulations. (c) Rotation angle β as a function of applied magnetic field strength up to 200 mT. (d) Example of folding Miura magneto-origami from the 2D magnetic sheet. in which red solid lines represent the mountain folds while black dashed lines denote valley folds. The left insert shows the magnetic polarity pattern where cross means magnetization pointing inward while blue represents magnetization pointing outward. The right insert shows the finalized Miura magneto-origami. (e) Demonstration of shape morphing of Miura magneto-origami under different magnetic fields. (f) Statistical analysis of reproducibility and performance evaluation of Miura magneto-origami fabricated by 8 volunteers each of whom folded 7 samples.

Supplementary Figure 8 A cuboid permanent magnet for actuating magneto-origami machines. (a) Schematic of the cuboid magnet with 25 mm in width, 50 mm in length, and 25 mm in height. (b) The magnetic field around the cuboid magnet. The distribution and magnetization of the permanent magnet was obtained using COMSOL Multiphysics software, under the same conditions including dimensions and magnetization. (c) Schematic of two parameters (distance d and angle α) to quantitatively control the permanent magnet. (d) The magnetic field strength as a function of distance d along the centerline of the magnet ($\alpha = 0^\circ$).

Supplementary Figure 9. Finite element analysis (FEA) of single-crease magneto-origami machine actuated by a cuboid permanent magnet. (a) FEA model of single-crease magneto-origami machines. (b) The FEA model for single-crease magneto-origami machine and detailed parameters of the model. (c) Meshed

FEA model in COMSOL. (d) Folding state of the single-fold model when the residual magnetization the permanent magnet is $1000 \text{ kA} \cdot \text{m}^{-1}$.

On Page 7, we add “**Shape-morphing mechanism of magneto-origami machines.** Fully magnetized magneto-origami machines retain the residual magnetic polarity patterns, i.e., magnetization \mathbf{M} . When subjected to an actuation magnetic field, magneto-origami machines can deform and move. In this work, we select a cuboid permanent magnet (NdFeB, dimension: $50 \text{ mm} \times 25 \text{ mm} \times 20 \text{ mm}$, residual magnetization 10^3 kA/m) as the magnetic actuation source (Supplementary Figure 8) because it is easier to handle and can generate stronger magnetic fields (up to 200 mT in this work) than electromagnetic coils. By varying the position/orientation of the cuboid magnet, the magnetic field strength \mathbf{B} around the magneto-origami machine can be effectively changed (Supplementary Figure 8). For example, the magnetic field strength on the centerline of the cuboid magnet can reach up to 200 mT when the distance between the center of the magnet and the magneto-origami machine is around 22.5 mm (Supplementary Figure 8).

To elucidate the magnetically controllable shape-morphing mechanism, here we adopt a single-crease magneto-origami machine as a representative example (Figure 2a). Upon actuation, the upper panel of the magneto-origami machine rotates around the crease due to the driving force, i.e., the magnetic torque density $\mathbf{M} \times \mathbf{B}$ and body force $(\nabla \mathbf{B})\mathbf{M}$ where operator “ \times ” and “ ∇ ” represents the cross-product and gradient, respectively. Therefore, the total magnetic torque that deforms the upper panel can be calculated as

$$\mathbf{T}_m = \int_{\Omega} [\mathbf{M} \times \mathbf{B} + \mathbf{r} \times (\nabla \mathbf{B})\mathbf{M}] dv \quad (1)$$

where Ω represents the 3D domain of the upper panel and \mathbf{r} is the position vector of a spatial point on the upper panel with respect to the crease (Figure 2a). According to the literature² and our experimental validation (Supplementary Figure 6), the folding crease behaves like an elastic hinge in which the total elastic torque is linearly proportional to angle change of the crease, i.e.,

$$\mathbf{T}_e = kL(\beta - \beta_0) \quad (2)$$

where $k = 1.47 \mu\text{N/deg}$ represents the experimentally measured torsional constant of the crease per unit width, L , $\beta_0 \approx 130^\circ$ and β denote the width, the rest angle, the deformed angle of single-crease magneto-origami, respectively (Figure 2a, Supplementary Figure 6). Therefore, the deformed shape of the magneto-origami machine can be found when the equilibrium is reached, i.e.,

$$\mathbf{T}_m = \mathbf{T}_e \quad (3)$$

Considering that magnetic fields around a cuboid magnet are nonuniform, Eq. (3) is solved by finite element analysis (FEA) (Supplementary Figure 9). The predicted folding angle β by FEA under various magnetic fields up to 200 mT are validated by experimental measurements (Figure 2b, 2c). Note that when the magnetic field strength is 200 mT , the magneto-origami is completed folded. Upon removal of the actuation field, the magneto-origami machine rapidly recovers the rest state. This reversible folding and unfolding process is shown in Movie 2.”

DISCUSSION

On Page 13, we add “Currently, magneto-origami machines are still in their infancy. Although substantial efforts have been made on designing folding patterns in literature, fabrication is still heavily associated with

manual folding. In this regard, we have done two reproducibility tests to evaluate the finished shape and the shape-morphing behavior of the magneto-origami machine. Future work can be focused on how to develop the automated line for folding origami with consistent shape and performance. In addition, actuating magneto-origami machines by the cuboid magnet is performed by human hands. More accurate control can be done by integrating the cuboid magnet with a robotic arm.”

Comment 3.

Can the spring actuator bend the shape without fixing the base (end of one side) on the floor? Maybe not. When the magnet moves as shown in Figure 4d, the spring actuator will be pulled to the magnet and will not make this motion. Detail explanation of the experimental condition for each example should be shown for proper evaluation of the methods.

Response 3.

The spring actuator can not bend without fixing the base. This condition has been explicitly expressed in the revised manuscript.

Also, we have added quantitative control schemes in the revised Figures 4, 5.

On Page 11, we add “Second, by fixing the base of the spring actuator, rotating the magnet around the spring actuator changes the magnetic field direction, yielding the bending configuration of the spring actuator as shown in Figure 5d.”

We added a quantitative control scheme in the revised Figure 4, 5 to help the reader understand the control of the magneto-origami strips.

On Page 9, we add “Corresponding quantitative control scheme by adjusting the position and rotation angle α of the magnet (see insert in Figure 3a-3c, Supplementary Figure 8c) while keeping the distance $d = 22.5$ mm ($B \sim 200$ mT) are provided for each strip.”... “The corresponding origami and magnetization pattern of each magneto-origami strip is shown below the experimental pictures. The quantitative control scheme is also provided.”

On Page 11, we add “We demonstrate such rolling locomotion by navigating the spring actuator through several obstructions including stairs, obstacles, narrow path, and groove by controlling the distance d , rotation angle α and position of the magnet in Figure 5f (Supplementary Movie 6).”

Revised Figure 4. Sequential folding and unfolding of the magneto-origami strips. (a-c) The schematic illustrations and experimental demonstrations of magneto-origami strips programmed with alternating magnetic polarity patterns. The magneto-origami strips can be sequentially folded into predefined 3D shapes: (a) lamellar, (b) triangular, and (c) square, by moving and rotating the permanent magnet. Quantitative control schemes are provided in which rotation angle α (left axis, blue curve) and distance d between the magnet and magneto-origami strip is plotted as a function of the magnet position. (d) Experimental demonstrations and quantitative control schemes of sequential folding of magneto-origami strips into letters “b”, “m”, and “e”.

Revised Figure 5. Actuations of magneto-origami spring actuator. (a) The spring actuator can contract when the magnet is placed in its axial direction. (b) The contraction ratio $(L_0 - L)/L_0$ of the spring actuator plotted as a function of magnetic field strength up to 220 mT. (c) The spring actuator remains 93% of the original length L_0 after a 1000-cycle actuation ($B \sim 220$ mT). (d) Bending behaviors of the spring actuator. Panels show bending angles are 0° , 45° , 90° , and 180° , respectively. (e) The 360° rotation behavior of a bent spring actuator by rotating the magnet surrounding the actuator. The yellow arrow represents the rotation direction of the rotating magnet. (f) Experimental demonstration of the rolling locomotion of the spring actuator by passing through several obstructions including stairs, obstacle, narrow path, and groove. The height of the stair, obstacle, and narrow path is 10 mm, 10 mm, and 12 mm, respectively. The width of the groove is 10 mm. Quantitative control schemes are provided in which rotation angle α (left axis, blue curve) and distance d between the magnet and magneto-origami strip is plotted as a function of the magnet position.

Comment 4.

What are the benefit or advantages of each demonstration in this study? In the reviewer's view, most of the demonstration can be achieved without the help of origami or like this complex design. A scientific or technological contribution should be addressed.

Response 4.

According to ref [Rus D & Tolley MT, Design, fabrication and control of origami robots. *Nature Reviews Materials* 3(6):101-112 (2018)], key benefits/advantages of origami include (1) shape-morphing between 2D and 3D configurations with large area change; (2) capability of the large degree of deformation (e.g., bending, stretching) to adapt their bodies to the task and environment; (3) simple/rapid design and fabrication method for customized structures and complex mechanisms. These advantages have been widely demonstrated in the literature.

In our work, we incorporate origami with magnetically controllable shape-morphing capability by simply replacing the raw paper with a magnetic sheet. Demonstrations are carefully selected to show the abovementioned advantages together with magnetic responsiveness.

- (1) The magneto-origami starshade (Figure 3) and magneto-origami charging robot (Figure 6a) exhibit 3D-to-2D shape-morphing with large area change (as inspired by the origami structures used in energy harvesting [Rösch AG, *et al.* Fully printed origami thermoelectric generators for energy-harvesting. *npj Flexible Electronics* 5, 1 (2021); Zhao L, *et al.* Shape-Programmable Interfacial Solar Evaporator with Salt-Precipitation Monitoring Function. *ACS Nano* 15, 5752-5761 (2021)]).
- (2) The magneto-origami strip actuator (Figure 5) and magneto-origami 8-3 encoder (Figure 6b) show high stretching ratio and large bending angle to adapt to task and environment (as inspired by the origami robotic arm [Wu S, *et al.* Stretchable origami robotic arm with omnidirectional bending and twisting. *PNAS* 118(36):e2110023118,(2021); Novelino LS *et al.*, Untethered control of functional origami microrobots with distributed actuation. *PNAS* 117(39):24096-24101, (2020)]).
- (3) The magneto-origami quadruped robot (Figure 6c) demonstrates a simple way to construct a robot with cargo transportation and release capability (as inspired by the origami robot [Miyashita S *et al.* An untethered miniature origami robot that self-folds, walks, swims, and degrades. In: *IEEE International Conference on Robotics and Automation (ICRA)*, IEEE (2015); Guitron S, Guha A, Li S, Rus D. Autonomous locomotion of a miniature, untethered origami robot using hall effect sensor-based magnetic localization. In: *IEEE international conference on robotics and automation (ICRA)*. IEEE (2017)]).

To alleviate concerns, we clearly describe the origami-enabled benefit/advantages of demonstration in the revised manuscript. We also provide further discussions in DISCUSSION.

On Page 9, we add “It is worth noting that both the folding of the magneto-origami square-twist and the unfolding of the magneto-origami starshades complete in a short time of 8 s with a change in the area by two folds. The large area change is intrinsically enabled by the origami structure and the on-demand shape-morphing is highly desired for applications such as energy harvesting^{39, 40}.”

On Page 12, we add “This demonstration clearly shows that utilizing the 3D-to-2D shape-morphing, the magneto-origami machine can have on-demand large area change.”

On Page 13, we add “The demonstration of the quadruped robot manifests that by combining different magneto-origami, we can construct magnet soft robots with advanced functionalities for complex tasks.”

Comment 5.

Quantification analysis (e.g. moving speed, magnetization property) compared to the relevant studies are requested for performance validation.

Response 5.

In **Response 1**, we have thoroughly discussed our novelty with existing works. Specifically for magnetization property, many existing works utilize laser programming, magnetic spray, etc, to encode the magnetic polarity, which is time-consuming. In our work, since making magneto-origami machines by folding the magnetic sheet, we can encode the magnetic polarity by the impulse magnetic field at one time, which is fast. The residual magnetization strength is 170 kA/m and the magnetization direction on different domains is intrinsically enabled by folding. We provide Supplementary Figure 1 to help understand. Please refer to our **Response 1** for details.

We also add quantification comparisons to the relevant studies on the moving speed and required magnetic field strength of the spring actuator (Figure 5). On Page 11, we add “The moving speed of the spring actuator is about 2 mm/s which is comparable to relevant studies (Supplementary Figure 14).”

Reference

- [1] T.Q. Xu, J.C. Zhang, M. Salehizadeh, O. Onaizah, E. Diller, Millimeter-scale flexible robots with programmable three-dimensional magnetization and motions, *Sci Robot* 4(29) (2019).
- [2] X. Yang, W.F. Shang, H.J. Lu, Y.T. Liu, L. Yang, R. Tan, X.Y. Wu, Y.J. Shen, An agglutinate magnetic spray transforms inanimate objects into millirobots for biomedical applications, *Sci Robot* 5(48) (2020).
- [3] Y. Li, Z.J. Qi, J.X. Yang, M.X. Zhou, X. Zhang, W. Ling, Y.Y. Zhang, Z.Y. Wu, H.J. Wang, B.A. Ning, H. Xu, W.X. Huo, X. Huang, Origami NdFeB Flexible Magnetic Membranes with Enhanced Magnetism and Programmable Sequences of Polarities, *Adv Funct Mater* 29(44) (2019).
- [4] W.Q. Hu, G.Z. Lum, M. Mastrangeli, M. Sitti, Small-scale soft-bodied robot with multimodal locomotion, *Nature* 554(7690) (2018) 81-85.
- [5] H.J. Lu, M. Zhang, Y.Y. Yang, Q. Huang, T. Fukuda, Z.K. Wang, Y.J. Shen, A bioinspired multilegged soft millirobot that functions in both dry and wet conditions, *Nat Commun* 9 (2018).
- [6] H. Ceylan, N.O. Dogan, I.C. Yasa, M.N. Musaoglu, Z.U. Kulali, M. Sitti, 3D printed personalized magnetic micromachines from patient blood-derived biomaterials, *Sci Adv* 7(36) (2021).
- [7] C. Li, G.C. Lau, H. Yuan, A. Aggarwal, V.L. Dominguez, S.P. Liu, H. Sai, L.C. Palmer, N.A. Sather, T.J. Pearson, D.E. Freedman, P.K. Amiri, M.O. de la Cruz, S.I. Stupp, Fast and programmable locomotion of hydrogel-metal hybrids under light and magnetic fields, *Sci Robot* 5(49) (2020).
- [8] J.E. Park, J. Jeon, J.H. Cho, S. Won, H.J. Jin, K.H. Lee, J.J. Wie, Magnetomotility of untethered helical soft robots, *Rsc Adv* 9(20) (2019) 11272-11280.

Supplementary Figure 14. Comparisons to the relevant studies on the (a) moving speed and (b) required magnetic field of the spring actuator.

Reviewer # 4

General comment.

This paper presents various performances of magnetically foldable and actuatable origami devices. The performances include actuation of simple deploying/folding structures, centimeter-sized shape-changing locomoting robots, and an electronics-equipped power transmitting origami sheet. Authors claim that due to the realization of “mass production method of the magneto-active machine(s) with customized architectures and advanced functionality”, whose process is called “2D-to-4D fabrication” presented in the paper, the approach shows advantages to other fabrication methods such as laser programming, 2D molding (I believe this is what the authors intend to say in Table 2), 3D printing, and assembly-based fabrication. This is an effort-intensive study presenting unique and reliable results. Many results are well achieved. I believe, however, with a few more tweaks detailed below and the paper could get better.

Response.

We sincerely thank the reviewer for your positive comments about our work.

Comment 1.

First, the technical challenge addressed in the paper is obscurely defined. If the main contribution is the method of automated production of magneto-active machines, the authors should more clearly show the limitations of the method with more clear presentations in the results and outcomes. I'd like to see a movie of what is presented in Figure 1 if the process is configured as depicted; e.g. the curing and the laser cutting are subsequently done. Also as this is a self-folding/unfolding study, what kind of shapes are self-foldable (can be automated in the fabrication) and not self-foldable with the proposed method, and more clear descriptions on the involvement of (human) processes in the fabrication and the control are required.

Response 1.

In this work, we empower the traditional paper origami with magnetically responsive functionality by simply replacing the raw paper with a paper-based magnetic sheet (revised Figure 1a). To strengthen our work, we have built a roll-to-roll platform to achieve an automated fabrication of magnetic sheets with desired patterns as shown in the revised Figure 1b-1e. The principle of designing magneto-origami patterns remains the same as that of traditional paper origami, which allows us to fabricate a broad range of 3D magneto-origami machines with high structural resolution and multimodal shape-morphing/locomotion modes as showcased in the revised Figure 1g.

In the revised manuscript, we have added Supplementary Table 1 and significantly modified **INTRODUCTION** to clearly describe the novelty of our work compared with state-of-the-art works. In summary, (1) from the materials perspective, we are the first to prepare the magnetic sheet by coating and curing a layer of magnetic composite, which allow us to directly create 3D magneto-origami machines with a high structural resolution by folding; (2) from fabrication perspective, we are the first to build the roll-to-roll platform, which paves the way for high-throughput fabrication; (3) from the functionality perspective, we demonstrate on-demand shape-morphing/locomotion of soft magneto-active machines with origami-enabled

advantages such as large deformation (stretch ratio, bending angle, etc) and high area changes (folding to unfolding and vice versa).

We also provided some discussion on the limitations of our work in **DISCUSSION**.

INTRODUCTION

On Page 2, we add “Despite various strategies for fabricating soft magneto-active machines have been proposed (e.g., template molding,¹² extrusion-based 3D printing,¹³⁻¹⁵ laser printing,^{16, 17} voxel assembly¹⁸, and transfer printing¹⁹), existing fabrication methods suffer from limitations of the low structural complexity and/or time-consuming process (Supplementary Table 1). For example, the template molding method may only make 2D structures^{12, 20} and the extrusion-based 3D printing fails to print complex structures due to the viscosity and the die swell of the composite ink.²¹ The laser programming method usually requires a relatively long time to encode magnetic polarity patterns in a large area.^{16, 22} Therefore, the fast fabrication of soft magneto-active machines with customized architectures yet remains an unresolved challenge.”

Supplementary Table 1. Comparison between this work and existing soft magneto-active machines.

Fabrication Method	Materials	Magnetization encoding	Structural complexity	Time cost	Production scale	Locomotion modes	Reference
3D printing	Hard magnetic particle @ polymer matrix	Sequential encoding while printing	Low	Intermediate	Intermediate	Bending/rolling /walking	(13-15)
Laser printing	Hard magnetic particle @ polymer matrix	Sequential encoding while heating	Low	High	Small	Bending/rolling /walking	(16-17)
2D template molding	Soft magnetic particle @ polymer matrix	Encoding at one time while curing	Low	High	Intermediate	Bending/Crawling	(12)
Voxel assembly	NdFeB @ PDMS voxel +Dragon skin voxel +Ecoflex bonding agent	Encoding before assembly	High	High	Small	Bending/twisting /contraction	(18)
Transfer printing	NdFeB @ PDMS +Stamp film +Adhesive	Encoding before assembly	Intermediate	High	Small	Bending/crawling	(19)
Origami +Assembly	Permanent magnets +Origami (e.g, polystyrene)	Permanent magnet orientation	High	Low	Intermediate	Contraction/deploying /bending/twisting	(29-33)
Origami +Magnetic Spray	Fe @ PVA&Gluten + Paper	Sequential encoding while curing	High	High	Small	Folding/rolling /walking/crawling	(35)
Origami	Magnetic sheet: NeFeB+ PEG shell @ Ecoflex	Encoding at one time while heating	Intermediate	High	Small	Bending/folding	(34)
Origami + Roll-to-roll	Magnetic sheet: NeFeB @ Ecoflex on paper	Encoding at one time after curing	High	Low	Large	Bending/folding/ contraction/ deploying/rolling	This work

On Page 3, we add “A simple way to empower origami with magnetic responsiveness is to directly adhere permanent magnets to the origami.²⁹ For example, Rus’s group has created a few magnet-based origami robots and demonstrated the capability of locomotion³⁰, metamorphosis,³¹ swimming,³² and drug delivery.³³ However, the presence of permanent magnets limits the softness and miniaturization of the robot, while increasing the undesired risk of magnet break-off in clinic applications.³⁴ Recently, Song *et al.* developed a soft magnetic sheet that can be folded into various magneto-origami structures³⁵. But without a backing layer,

such a pure magnetic sheet exhibits an inferior structural resolution to its counterpart of paper origami, especially at folding creases.³⁵ Yang *et al.* directly sprayed uncured magnetic composite to the origami, yet still experienced prolonged time in sequentially curing/magnetizing each part of the origami.³⁶ So far, to the best of our knowledge, no fabrication method leverages structural complexity, softness, resolution, and fast production of magneto-origami machines at a large scale.

Here, we propose a facile fabrication strategy that can fast construct soft magneto-active machines at a large scale by incorporating roll-to-roll processing of 2D patterns and 3D origami folding (Figure 1). By simply coating and curing a layer of magnetic composite (i.e., dispersing hard-magnetic microparticles in the polymer matrix) on a piece of raw paper, we create a paper-like soft magnetic sheet that can be directly folded into magneto-origami machines with customized geometries and high resolution. The magnetic polarity of the magneto-origami machine is quickly encoded at one time by a strong impulse magnetizing field ($H \sim 3T$). When subjected to an actuation magnetic field, the magneto-origami machine can achieve on-demand shape-morphing. We demonstrate a set of magneto-origami machines that can fold, bend, roll and walk with potential applications of deployment, object manipulation, and locomotion. Compared with existing soft magneto-active machines, our roll-to-roll platform allows for the automated fabrication of 2D magnetic patterns, followed by a quick magnetization process and direct folding, giving rise to facile and high-throughput production of magneto-active machines in various geometries and multimodal shape-morphing capabilities (Supplementary Table 1)."

RESULTS

On Page 4, we add "**Fabrication of soft magneto-origami machines.** The magnetic composite consists of magnetizable neodymium-iron-boron (NdFeB) particles (average diameter $\sim 30 \mu\text{m}$) embedded in the polymer resin (i.e., Ecoflex 00-10, Smooth-On) with a particle volume fraction of 25%. The 2D magnetic sheet is made by coating and curing a layer of magnetic composite on a piece of raw paper (Figure 1a, Supplementary Figure 1). To achieve an automated fabrication of 2D magnetic sheets, we build a roll-to-roll platform as shown in Figure 1b-1e. The major components of roll-to-roll platform consist of a roller of raw paper, a scraper (Figure 1c) that creates an even layer of uncured magnetic composite on the raw paper, a heater (Figure 1e) that cures the magnetic composite layer, a laser machine (Figure 1d) that cuts the magnetic sheet into designed patterns, and an end roller. After curing, the designed 2D pattern is cut by a laser machine with uncut connections to allow for an easy tear-off (Supplementary Figure 1, Supplementary Movie 1). The rolling velocity, the temperature of the heater, and the pattern of the magnetic sheet in the roll-to-roll process can be readily adjusted in an automated manner, which paves the way for the high-throughput fabrication of magneto-origami machines. Supplementary Movie 1 shows the detailed roll-to-roll process in which the production rate is 100 mm/min."

On Page 5, we add "To maintain the desired shape of the magneto-origami machine (e.g., to avoid shape collapse or excessive recovery), the mechanical properties of the magnetic sheet are carefully designed. Different recipes of magnetic sheets are tested by coating magnetic composite on aluminum (Al) foil, and polyimide (PI) membrane, and raw paper (Supplementary Figure 3). It is found that the Al-based magnetic sheet is too floppy to sustain the shape well (similar to the backing-free magnetic sheet developed by Song et

al.³⁵), while the PI-based magnetic sheet is too stiff to be folded. By coating a 100 μm -thick magnetic composite layer on top of a 90 μm -thick raw paper (Figure 1f, Supplementary Figure 4), the paper-based magnetic sheet (Young's modulus ~ 7 MPa, ultimate tensile stress ~ 14 MPa, Supplementary Figure 5) preserves the excellent folding capability of paper, allowing for consistent fabrication of magneto-origami machines in a broad spectrum of geometries with a high structural resolution, as showcased in Figure 1g.”

Revised Figure 1. Fabrication of magneto-origami machines. (a) Schematic illustration of fabrication processes of the patterned magnetic sheet by roll-to-roll method, folding/magnetizing, and partial recovery.

(b)-(e) Experimental pictures of (b) laser cutting, (c) scraper, (d) pattern, (e) curing heater of the roll-to-roll platform to fabricate patterned magnetic sheets. (f) Images showing a roll of raw paper and magnetic sheet with a 2D origami pattern. (g) Various magneto-origami machines with customized geometries and high structural resolution can be consistently fabricated.

DISCUSSION

On Page 13, we add “Origami, originating from the ancient Asia art of paper, has nowadays been extensively explored as a fabrication strategy of engineering structures via a sequence of spatially organized folds. The key benefit of origami fabrication is the simple design principle and efficient creation of complex structures with customized geometries by constructing patterns in a plane and folding them into their final shape.²⁵ When utilized to design robots, origami robots exhibit intrinsic shape-morphing capability between 2D and 3D configurations. Enabled by multi-folds, the shape-morphing can have a large degree of deformation (e.g., stretching, contraction, bending), adapting the origami robots to different tasks and environments. In this work, we transform the conventional paper into a magnetic sheet by coating a thin layer of magnetic composite on the paper. The magnetic sheet preserves the paper-like foldability while imparting magnetic responsiveness to the magneto-origami machines it creates. We have built a roll-to-roll platform that achieves an automated fabrication of 2D magnetic patterns and demonstrate several 3D magneto-origami machines that can change their morphologies and execute locomotion tasks on demand, e.g., deploying, locking, sequential folding, and cargo transportation.

Currently, magneto-origami machines are still in their infancy. Although substantial efforts have been made on designing folding patterns in literature, fabrication is still heavily associated with manual folding. In this regard, we have done two reproducibility tests to evaluate the finished shape and the shape-morphing behavior of the magneto-origami machine. Future work can be focused on how to develop the automated line for folding origami with consistent shape and performance. In addition, actuating magneto-origami machines by the cuboid magnet is performed by human hands. More accurate control can be done by integrating the cuboid magnet with a robotic arm. Furthermore, the magnetization of magneto-origami machines can also be encoded before they are folded, giving rise to different magnetic polarity patterns and shape-morphing behavior of the machine (Supplementary Figure 16, Supplementary Movie 10)”

Comment 2.

Second, many of these performances such as deployable structures, sequential folding structures, a locomoting robot, wireless powering sheet, lack comparisons on their performances with the other works. Only two (review) papers on origami robots were referenced (line 48, refs 20 and 21) in this regard, making the comparison of results with other similar robotics performances difficult. For example, the authors claim that “The deployable magneto-origami electronic robot can unfurl by 12 times of its folded area..(line 218)”. The achievement of 12 times is owing to the crease pattern invented by another person, and the impact of 12 times is difficult to assess in the context. While it is meaningful to see that the proposed method can be applied to different performances and applications, the presentation style of results gives an impression that the performances were heuristically sought and designed, and are not necessarily coherent. Limited use of passive voices in the text with a clearer presentation of the subjects should be considered (e.g. line 73: .. the torn pattern is folded into a desired 3D origami.., line 110 .. magneto-origami flower is constructed..; line 229 .. ink was uniformly coated..).

Given that the basic principle of “Programming magnetic anisotropy in polymeric microactuators” by Kim et al., <https://www.nature.com/articles/nmat3090> (2011) (which is missing in the reference) which was further advanced in (13) or (17), and therefore a contribution of the work is limited to the semi-automated fabrication method and the versatile applicability of the method in different performances, higher theoretical solidity in these aspects are wanted.

Response 2.

Thank you for your comment. We have added more references to the revised manuscript. Please refer to our **Response 1** for detailed comparisons with existing works (**Supplementary Figure 1**). We have deleted the claim about “automated fabrication of magneto-origami machines”. Since we have built a roll-to-roll platform, we now only claim “automated fabrication of magnetic sheet”.

We also added theoretical and numerical analysis to further strengthen the theoretical solidity of our work. In the Discussions, we added some discussions on the limitations of current work and future perspectives. We have revised/deleted those ambitious descriptions as suggested.

Revised Figure 2. Theoretical analysis, finite element simulations, and reproductivity test of magneto-origami machines. (a) Schematic illustration showing the shape-morphing of the single-crease magneto-origami actuated by a magnetic field. Equilibrium is reached when the total driving magnetic torque T_m is equal to elastic torque T_e . (b) Comparison of deformation of single-crease origami between experiments and finite element simulations. (c) Rotation angle β as a function of applied magnetic field strength up to 200 mT. (d) Example of folding Miura magneto-origami from the 2D magnetic sheet. In which red solid lines represent the mountain folds while black dashed lines denote valley folds. The left insert shows the magnetic polarity pattern where cross means magnetization pointing inward while blue represents magnetization pointing outward. The right insert shows the finalized Miura magneto-origami. (e) Demonstration of shape morphing of Miura magneto-origami under different magnetic fields. (f) Statistical analysis of reproductivity

and performance evaluation of Miura magneto-origami fabricated by 8 volunteers each of whom folded 7 samples.

Supplementary Figure 8 A cuboid permanent magnet for actuating magneto-origami machines. (a) Schematic of the cuboid magnet with 25 mm in width, 50 mm in length, and 25 mm in height. (b) The magnetic field around the cuboid magnet. The distribution and magnetization of the permanent magnet was obtained using COMSOL Multiphysics software, under the same conditions including dimensions and magnetization. (c) Schematic of two parameters (distance d and angle α) to quantitatively control the permanent magnet. (d) The magnetic field strength as a function of distance d along the centerline of the magnet ($\alpha = 0^\circ$).

Supplementary Figure 9. Finite element analysis (FEA) of single-crease magneto-origami machine actuated by a cuboid permanent magnet. (a) FEA model of single-crease magneto-origami machines. (b) The FEA model for single-crease magneto-origami machine and detailed parameters of the model. (c) Meshed FEA model in COMSOL. (d) Folding state of the single-fold model when the residual magnetization the permanent magnet is $1000 \text{ kA} \cdot \text{m}^{-1}$.

On Page 7, we add “**Shape-morphing mechanism of magneto-origami machines.** Fully magnetized magneto-origami machines retain the residual magnetic polarity patterns, i.e., magnetization \mathbf{M} . When subjected to an actuation magnetic field, magneto-origami machines can deform and move. In this work, we select a cuboid permanent magnet (NdFeB, dimension: $50 \text{ mm} \times 25 \text{ mm} \times 20 \text{ mm}$, residual magnetization 10^3 kA/m) as the magnetic actuation source (Supplementary Figure 8) because it is easier to handle and can generate stronger magnetic fields (up to 200 mT in this work) than electromagnetic coils. By varying the position/orientation of the cuboid magnet, the magnetic field strength \mathbf{B} around the magneto-origami machine can be effectively changed (Supplementary Figure 8). For example, the magnetic field strength on the centerline of the cuboid magnet can reach up to 200 mT when the distance between the center of the magnet and the magneto-origami machine is around 22.5 mm (Supplementary Figure 8).

To elucidate the magnetically controllable shape-morphing mechanism, here we adopt a single-crease magneto-origami machine as a representative example (Figure 2a). Upon actuation, the upper panel of the magneto-origami machine rotates around the crease due to the driving force, i.e., the magnetic torque density $\mathbf{M} \times \mathbf{B}$ and body force $(\nabla \mathbf{B})\mathbf{M}$ where operator “ \times ” and “ ∇ ” represents the cross-product and gradient, respectively. Therefore, the total magnetic torque that deforms the upper panel can be calculated as

$$\mathbf{T}_m = \int_{\Omega} [\mathbf{M} \times \mathbf{B} + \mathbf{r} \times (\nabla \mathbf{B})\mathbf{M}] dv \quad (1)$$

where Ω represents the 3D domain of the upper panel and \mathbf{r} is the position vector of a spatial point on the upper panel with respect to the crease (Figure 2a). According to the literature² and our experimental validation (Supplementary Figure 6), the folding crease behaves like an elastic hinge in which the total elastic torque is linearly proportional to angle change of the crease, i.e.,

$$\mathbf{T}_e = kL(\beta - \beta_0) \quad (2)$$

where $k = 1.47 \mu\text{N/deg}$ represents the experimentally measured torsional constant of the crease per unit width, L , $\beta_0 \approx 130^\circ$ and β denote the width, the rest angle, and the deformed angle of single-crease magneto-origami, respectively (Figure 2a, Supplementary Figure 6). Therefore, the deformed shape of the magneto-origami machine can be found when the equilibrium is reached, i.e.,

$$\mathbf{T}_m = \mathbf{T}_e \quad (3)$$

Considering that magnetic fields around a cuboid magnet are nonuniform, Eq. (3) is solved by finite element analysis (FEA) (Supplementary Figure 9). The predicted folding angle β by FEA under various magnetic fields up to 200 mT are validated by experimental measurements (Figure 2b-c). Note that when the magnetic field strength is 200 mT , the magneto-origami is completed folded. Upon removal of the actuation field, the magneto-origami machine rapidly recovers the rest state. This reversible folding and unfolding process is shown in Movie 2.”

Comment 3.

- (1) Line 54: I'm not sure if reference 27 should appear here in the context of origami structure.
- (2) Line 75: application of impulsive magnetic field for the wireless reprogramming of a magnetic (NdFeB) structure was presented by Sitti et al. "Remotely Addressable Magnetic Composite Micropumps. RSC Advances, 2012". Authors often use the words "direct 2D-to-4D fabrication". What do you mean by "direct" here?
- (3) Line 94: "... unfolding process is shown in..". The crease is only partially unfolded (and folded). How does the motion of the cuboid magnet affect the folding angle?
- (4) Figure 3b and Figure 3c: It is not clear in the text how the magnet was moved to form different shapes. Can you mathematically formulate in the supporting material?
- (5) Line 170: "... we develop a deployable magneto-origami electronic robot..". There are planar origami structures that can be empowered with magnetic induction (e.g. Rob Wood from Harvard microrobotics lab), even the same setup may not exist, some references to the previous research are needed.
- (6) Line 176: The authors claim that they demonstrated "electrical stimulation therapy". However, I don't see the stimulation performance. I assume the alternative magnetic field for magnetic induction was managed by a low-inductance coil which is not described in the paper.

Response 3.

- (1) Reference 27 has been removed.
- (2) Reference is cited now. We have removed the claim "direct 2D-to-4D" in the revised manuscript.
- (3) We have added theoretical analysis, numerical simulation, experimental validations in the revised Figure 2. Please refer to our **Response 2** (Figure 2c in particular) for details.
- (4) The quantitative control schemes are provided in Figure 3 (now Figure 4 on next page) in which the distance between the magnet and the magneto-origami machine and the rotation angle is plotted against the position. We also add descriptions. On Page 9, we add "Corresponding quantitative control scheme by adjusting the position and rotation angle \$\alpha\$ of the magnet (see insert in Figure 3a-3c, Supplementary Figure 8c) while keeping the distance \$d = 22.5\$ mm (\$B \sim 200\$ mT) are provided for each strip."... "The corresponding origami and magnetization pattern of each magneto-origami strip is shown below the experimental pictures. The quantitative control scheme is also provided."
- (5) Reference [Boyvat M, Koh J-S, Wood RJ. Addressable wireless actuation for multijoint folding robots and devices. Science Robotics 2, eaan1544 (2017)] has been cited as ref 45.
- (6) We have removed such a claim.

Revised Figure 4. Sequential folding and unfolding of the magneto-origami strips. (a-c) The schematic illustrations and experimental demonstrations of magneto-origami strips programmed with alternating magnetic polarity patterns. The magneto-origami strips can be sequentially folded into predefined 3D shapes: (a) lamellar, (b) triangular, and (c) square, by moving and rotating the permanent magnet. Quantitative control schemes are provided in which rotation angle α (left axis, blue curve) and distance d between the magnet and magneto-origami strip is plotted as a function of the magnet position. (d) Experimental demonstrations and quantitative control schemes of sequential folding of magneto-origami strips into letters “b”, “m”, and “e”.

REVIEWER COMMENTS

Reviewer #1 (Remarks to the Author):

The authors have substantially revised their manuscript and the content is now much improved. I could recommend it for publication given that the authors address the following concerns:

1. From Fig. 2e, it can be clearly seen that this paper-based origami structure is non-rigid, which means that the panels bend and stretch significantly during folding. This may affect the expected foldability of origami patterns, leading to folding sequences that cannot be simply predicted by geometry and kinematics. Please comment on how this phenomenon could be predicted and plans to address this issue, such as performing reduced order modeling, or increase panel stiffness.

2. Response 5 to reviewer #1 is not particularly convincing. In this work, the authors only tested simple patterns that involves trivial motions. Perhaps it is more accurate to say that "we conclude that failure rate in current work is negligibly small for commonly seen origami patterns to perform simple motions." Please included failure rate related comments also in the main text.

3. I appreciate the new experiments done by the authors to answer comment 9 of reviewer 1. However, the conclusion is incorrect, given the data shown by the authors. According to the force-extension plot in Supplementary Figure 12b, none of the samples is displaying bistability. Force is the gradient of energy w.r.t to extension. Therefore, if there is no negative force appearing, the energy of the system is still monotonically increasing, that is, there is no local minimum other than the initial configuration. If the authors carefully check Figure R1, there is a region that the force curve goes under 0. A possible reason for not seeing bistable behavior could be that the stiffness of the creases is relatively high. I am not expecting the authors to redo the experiments, but this wrong conclusion of "bistability is observed" must be corrected.

Reviewer #2 (Remarks to the Author):

I appreciate the authors' efforts to improve the manuscript, and think overall my concerns have been alleviated. In particular, the inclusion of a roll-to-roll manufacturing method is excellent. I have a few minor comments that may improve the paper, below.

- Movie 9 - I like the addition of the single-leg robot. Suggest putting the magnet position and orientation to match the 4-leg robot video (also Movie 9).

- Manuscript - suggest copy-editing to watch verb tenses etc.

For example, in introduction:

-- "Despite various strategies for fabricating soft magneto-active machines have been proposed (e.g., template molding,⁴⁶ 12 extrusion-based 3D printing,¹³⁻¹⁵ laser printing,^{16,17} voxel assembly¹⁸, and transfer printing^{19,47}), existing fabrication methods suffer from 48 limitations of the low structural complexity and/or time-consuming process (Supplementary Table 1)."

-- Should be more like "Although various strategies for fabricating soft magneto-active machines have been proposed (~same~) many existing fabrication methods either can only achieve low structural complexity, or they are a time-consuming process."

Right before discussion: "But it falls when unloading the ball at the destination"  "It can carry...attraction, but it cannot unload the ball at the destination". Or just replace "falls" with "fails"

- Figure 2f - what is plotted? It is unclear. The legend is implicitly showing each of the 8 volunteers' results. I suggest improving the legend or caption, and explaining what the error bars are - 1 SD? 95% CI? Be specific.

- Small note - Figure 5. If you can make the control scheme have the same width as the traversed terrain, that would make it clearer. I understand there are competing objectives - make photo large, vs. make photo align with control scheme. Just a comment/thing to think about.

- I like the inclusion of the basic theory in eq. 1-3. If the authors could plot the analytical alongside the FEA, that would be ideal, but not strictly "necessary".

Reviewer #3 (Remarks to the Author):

Supplementary demonstrations are attractive and interesting. However, technological advancement is still curious on the aspect of recent advancement of magneto-responsive machines or robots. Moreover, the reviewer's concern on the previous manuscript was not clearly resolved while revision, especially for the control and related demonstration. Specifically, the reviewer still wonders how the authors can make the precise movement of the permanent magnet continuously by maintaining the distance and angle between the object and the magnet. It is not shown in the supplementary video clip (simply depicted as an animation), that should be shown in real-time movement. Overall, the contribution of the paper is limited to the fabrication method of magneto-responsive sheet, but the core scientific finding or methodological advancement are not significant.

Reviewer # 1

General Comment. The authors have substantially revised their manuscript and the content is now much improved. I could recommend it for publication given that the authors address the following concerns:

Comment 1.

From Fig. 2e, it can be clearly seen that this paper-based origami structure is non-rigid, which means that the panels bend and stretch significantly during folding. This may affect the expected foldability of origami patterns, leading to folding sequences that cannot be simply predicted by geometry and kinematics. Please comment on how this phenomenon could be predicted and plans to address this issue, such as performing reduced order modeling, or increase panel stiffness.

Response 1.

Figure R3 Explanation of model of origami folding. (a) The top panel is assumed as rigid while the folding crease is modeled as an elastic hinge. (b) Experiments validate the top panel remains almost flat during folding/unfolding even after 1000 cycles (adopted from Supplementary Figure 7 and 10).

We thank the reviewer for commenting on our work again. We clarify that the adjective “rigid” is used to approximately describe the top panel rather than the entire origami structure (Figure R3 a). In other words, the top panel remains flat during the folding/unfolding of the origami. The restored strain energy only comes from the deformed folding crease that is modeled as an elastic hinge in this work (see Eq. (2)). The shrinkage of the Miura magneto-origami machine in original Figure 2e (now Figure 2h) is mainly due to the folding of the crease while (top) panels do not bend or stretch. This assumption is widely adopted in literature when modeling origami structures (e.g., Nat. Mat. 14, 4232, 2015; Nat. Phys. 14,811-815,2018;). Also, we validate this assumption by experiments (Figure R3 b) that the top panel remains flat during folding/unfolding even after 1000 cycles.

Comment 2.

Response 5 to reviewer #1 is not particularly convincing. In this work, the authors only tested simple patterns that involves trivial motions. Perhaps it is more accurate to say that “we conclude that failure rate in current work is negligibly small for commonly seen origami patterns to perform simple motions.” Please included failure rate related comments also in the main text.

Response 2.

We have adopted your suggestion.

On Page 16, we say “In the current work, the failure rate is negligibly small for commonly seen origami patterns to perform simple motions.”

Comment 3.

I appreciate the new experiments done by the authors to answer comment 9 of reviewer 1. However, the conclusion is incorrect, given the data shown by the authors. According to the force-extension plot in Supplementary Figure 12b, none of the samples is displaying bistability. Force is the gradient of energy w.r.t to extension. Therefore, if there is no negative force appearing, the energy of the system is still monotonically increasing, that is, there is no local minimum other than the initial configuration. If the authors carefully check Figure R1, there is a region that the force curve goes under 0. A possible reason for not seeing bistable behavior could be that the stiffness of the creases is relatively high. I am not expecting the authors to redo the experiments, but this wrong conclusion of “bistability is observed” must be corrected.

Response 3.

Thank you for your analysis and we have removed the “bistability” statements in the revised manuscript.

Reviewer # 2

General Comment. I appreciate the authors' efforts to improve the manuscript, and think overall my concerns have been alleviated. In particular, the inclusion of a roll-to-roll manufacturing method is excellent. I have a few minor comments that may improve the paper, below.

Comment 1.

- Movie 9 - I like the addition of the single-leg robot. Suggest putting the magnet position and orientation to match the 4-leg robot video (also Movie 9).

Response 1.

Thank you for your suggestion. We have added the magnet position and orientation are for single-leg robot in Supplementary Movie 10.

Comment 2.

- Manuscript - suggest copy-editing to watch verb tenses etc.

For example, in introduction:

-- "Despite various strategies for fabricating soft magneto-active machines have been proposed (e.g., template molding,⁴⁶ 12 extrusion-based 3D printing,¹³⁻¹⁵ laser printing,^{16,17} voxel assembly¹⁸, and transfer printing^{19,47}), existing fabrication methods suffer from 48 limitations of the low structural complexity and/or time-consuming process (Supplementary Table 1)."

-- Should be more like "Although various strategies for fabricating soft magneto-active machines have been proposed (~same~) many existing fabrication methods either can only achieve low structural complexity, or they are a time-consuming process."

- Right before discussion: "But it falls when unloading the ball at the destination"  "It can carry...attraction, but it cannot unload the ball at the destination". Or just replace "falls" with "fails"

Response 2.

Thank you for your careful reading. Your suggestions are adopted in the revised manuscript.

Comment 3.

- Figure 2f - what is plotted? It is unclear. The legend is implicitly showing each of the 8 volunteers' results. I suggest improving the legend or caption, and explaining what the error bars are? Be specific.

- Small note - Figure 5. If you can make the control scheme have the same width as the traversed terrain, that would make it clearer. I understand there are competing objectives - make photo large, vs. make photo align with control scheme. Just a comment/thing to think about.

Response 3.

Figure 2 has been updated. The revised caption of Figure 2j (original Figure 2f) now is (j) Statistical analysis of reproductivity and performance evaluation of Miura magneto-origami fabricated by 8 volunteers each of whom folded 7 samples. Each solid curve represents the mean width of 7 samples of a specific volunteer by manually operating the magnet. The error bar shows the width variation of the 7 samples. The dashed black curve shows the mean width of 7 samples fabricated by volunteer #8 operated by the robotic arm.

Figure 2. Theoretical analysis, finite element simulations, and reproductivity test of magneto-origami

machines. (a) Schematic illustration showing the shape-morphing of the single-crease magneto-origami actuated by a magnetic field. Equilibrium is reached when the total driving magnetic torque T_m is equal to elastic torque T_e . (b) Comparison of deformation of single-crease origami between experiments and finite element simulations. (c) Rotation angle β as a function of applied magnetic field strength up to 200 mT. (d) Example of folding Miura magneto-origami from the 2D magnetic sheet, in which red solid lines represent the mountain folds while black dashed lines denote valley folds. The left insert shows the magnetic polarity pattern where cross means magnetization pointing inward while blue represents magnetization pointing outward. The right insert shows the finalized Miura magneto-origami. (e) Manipulating Miura magneto-origami machine by a robotic arm. (f) By changing the actuation distance (denoted as d) with a robotic arm, the magnetic fields at the Miura magneto-origami machine can be accurately tuned. (g) Reducing the actuation distance d , the Miura magneto-origami machine shrinks. (h) Demonstration of shape-morphing of Miura magneto-origami under different magnetic fields. (i) Comparison between the measured magnetic field strength between the robotic arm and human hands of 8 volunteers at actuation distance $d=32.2$ and 43.0 mm, respectively. (j) Statistical analysis of reproductivity and performance evaluation of Miura magneto-origami fabricated by 8 volunteers each of whom folded 7 samples. Each solid curve represents the mean width of 7 samples of a specific volunteer by manually operating the magnet. The error bar shows the width variation of the 7 samples. The dashed black curve shows the mean width of 7 samples fabricated by volunteer #7 operated by the robotic arm.

Revised Figure 5 with the control scheme expanded.

The control scheme of Figure 5 has been made the same width as the traversed terrain.

Comment 4.

- I like the inclusion of the basic theory in eq. 1-3. If the authors could plot the analytical alongside the FEA, that would be ideal, but not strictly "necessary".

Response 4.

Thank you for your suggestion. Since the magnetic fields around the cubic magnet are nonuniform, it is not easy to calculate the analytical solution. We will not include it in the paper.

Reviewer # 3

Comment. Supplementary demonstrations are attractive and interesting. However, technological advancement is still curious on the aspect of recent advancement of magneto-responsive machines or robots. Moreover, the reviewer's concern on the previous manuscript was not clearly resolved while revision, especially for the control and related demonstration. Specifically, the reviewer still wonders how the authors can make the precise movement of the permanent magnet continuously by maintaining the distance and angle between the object and the magnet. It is not shown in the supplementary video clip (simply depicted as an animation), that should be shown in real-time movement. Overall, the contribution of the paper is limited to the fabrication method of magneto-responsive sheet, but the core scientific finding or methodological advancement are not significant.

Response.

We thank the reviewer for commenting on our work again. To alleviate your concerns, we have significantly revised our manuscript. First, we purchased a robotic arm (myCobot Pro 600, Elephant Robotics Co., Ltd., China) and integrate the cuboid magnet into it. Via digital control, we have improved the manipulation stability, accuracy and reproductivity. Figure 2 now is revised to include the manipulation of the magnet by the robotic arm. The robotic arm has been used for some regular modes of manipulations, such as translation, circulation, and rotation as shown in Supplementary Figure 12. Also, we compared the operation accuracy between human hands and robot arm and verified the small variance between them.

On Page 9, we add “In addition, we also evaluate the deformed shapes of Miura magneto-origami machines folded by different volunteers in magnetic fields. We first employ a robotic arm (Figure 2e, myCobot Pro 600, Elephant Robotics Co., Ltd., China) to precisely control the actuation distance (denoted as \$d\$ in Figure 2f). By varying \$d\$, the magnetic field strength can be effectively tuned (e.g., \$B=50\$ mT at \$d=43.0\$ mm, \$B=100\$ mT at \$d=32.2\$ mm, Supplementary Figure 8) to change the width of Miura magneto-origami machines (Figure 2g and 2h, Supplementary Movie 3). Statistical studies only show a small variation between all samples in the deformed width and excellent reversibility when the magnetic field is removed (Supplementary Figure 11), manifesting that the proposed fabrication method by roll-to-roll process and manual folding can construct magneto-origami machines with consistent geometries and shape-morphing performance.”

It is also worth noting that the robot arm may only be able to complete regular modes of manipulation such as translation, circulation, and rotation (Supplementary Figure 12). In this regard, manual operation by human hands may supplement some complex modes of manipulation. To verify the accuracy of manual operation, we compare both the measured magnetic field strength and deformed shapes of Miura magneto-origami machines operated by the robotic arm and human hands. Figure 2i shows that there is a small difference in the measured magnetic field strength when tuning the actuation distance $d=32.2$ mm ($B=100$ mT) and $d=43.0$ mm ($B=50$ mT) by robotic arm and hands of 8 volunteers. Besides, 8 volunteers manually tuned the actuation distance and recorded the mean width of 7 samples by themselves in Figure 2j (solid curves). These results are very close to that by robotic arm manipulation (dashed black curve in Figure 2j, Supplementary Figure 11), suggesting that manual manipulation of Miura-magneto origami machines is also reliable.”

Figure 2. Theoretical analysis, finite element simulations, and reproductivity test of magneto-origami machines. (a) Schematic illustration showing the shape-morphing of the single-crease magneto-origami actuated by a magnetic field. Equilibrium is reached when the total driving magnetic torque T_m is equal to elastic torque T_e . (b) Comparison of deformation of single-crease origami between experiments and finite element simulations. (c) Rotation angle β as a function of applied magnetic field strength up to 200 mT. (d) Example of folding Miura magneto-origami from the 2D magnetic sheet, in which red solid lines represent the mountain folds while black dashed lines denote valley folds. The left insert shows the magnetic polarity pattern where cross means magnetization pointing inward while blue represents magnetization pointing outward. The right insert shows the finalized Miura magneto-origami. (e) Manipulating Miura magneto-origami machine by a robotic arm. (f) By changing the actuation distance (denoted as d) with a robotic arm, the magnetic fields at the Miura magneto-origami machine can be accurately tuned. (g) Reducing the actuation distance d , the Miura magneto-origami machine shrinks. (h) Demonstration of shape-morphing of Miura magneto-origami under different magnetic fields. (i) Comparison between the measured magnetic field strength between the robotic arm and human hands of 8 volunteers at actuation distance $d=32.2$ and 43.0 mm, respectively. (j) Statistical analysis of reproductivity and performance evaluation of Miura magneto-origami fabricated by 8 volunteers each of whom folded 7 samples. Each solid curve represents the mean width of 7 samples of a specific volunteer by manually operating the magnet. The error bar shows the width variation of the 7 samples. The dashed black curve shows the mean width of 7 samples fabricated by volunteer #7 operated by the robotic arm.

Supplementary Figure 12. Demonstration of three regular modes of manipulation by robot arm. (a) Translation mode. (b) Circulation mode. (c) Rotation mode.

Second, we add one more interesting application of magneto-origami machines in the revised manuscript: **bionic magneto-origami arts/crafts.**

On Page 15, we add “**Bionic magneto-origami arts/crafts.** Enabled by the intrinsic folding capability of the magnetic sheet, magneto-origami machines can be built into magnetically responsive arts/crafts. By printing colorful pictures on the top paper, we can further create bionic magneto-origami arts/crafts. As a representative demonstration, we present a magneto-origami art – dancing butterfly and blooming flower – in Figure 6i to 6k. The designs of the bionic magneto-origami butterfly and flower are shown in Supplementary Figure 19. By controlling the magnet via the motor, the bionic magneto-origami butterfly flaps its wings while the flower is blooming. Background music can be played during the actuation process, leading to a beautiful and euphonic bionic art (Supplementary Movie 3). We envisage that one can manually control the motor to trigger the shape-morphing of the bionic magneto-origami machines, enjoying the fun of the hands-on operation and the beauty of the vivid art. Serving as a fascinating media for the dissemination of science and art, bionic magneto-origami arts/crafts may have a far-reaching educational impact on public audiences, especially kids and students if exhibited in museums and schools”

Revised Figure 6i-6k Demonstration of bionic magneto-origami arts/crafts– dancing butterfly and blooming

flower. (i) By controlling the magnet via the motor, the bionic butterfly can flap its wings and the flower is blooming, while beautiful background music can be played during the actuation process. (j) Top view of the bionic magneto-origami art. (c) Shape-morphing of the bionic flower and butterfly.

Supplementary Figure 19. Design of bionic magneto-origami machines. (a) Top view of the paper patagium of the bionic butterfly. (b) Bottom view of the magnetic composite of the bionic butterfly. (c) Magnetization profile of the bionic magneto-origami butterfly. (d) Top view of the paper patagium of the bionic flower. (e) Bottom view of the magnetic composite of the bionic flower. (f) Magnetization profile of the bionic magneto-origami flower.

We hope these key revisions have resolved your concerns. Together with the first-reported roll-to-roll method for high-throughput fabrication, we believe our work will pave a way for the future application of soft magnetic robots.

REVIEWERS' COMMENTS

Reviewer #1 (Remarks to the Author):

The authors have addressed my comments, and I am glad to recommend the manuscript for publication.

Reviewer #2 (Remarks to the Author):

The authors have made a solid manuscript, and responded adequately to the comments from the other esteemed reviewers. I have no further substantial comments.

The authors' changes and response to Reviewer 2 comment 1, 2, 4 are appropriate. I think there was miscommunication about Figure 5 f (comment 3).

What I was suggesting was to make the x axis of the plow exactly align with the position in the image. This would require more nuanced/thoughtful editing than just widening the plot, since the robot motion doesn't end exactly at the right side of the image. Either re-cropping the figure or changing the x-axis of the plot to start a little negative and end slightly after 200, so that the leftmost robot image is at $x=0$, and rightmost robot image is at $x=200$.

Response to Reviewer #2:

Comments:

The authors have made a solid manuscript, and responded adequately to the comments from the other esteemed reviewers. I have no further substantial comments.

The authors' changes and response to Reviewer 2 comment 1, 2, 4 are appropriate. I think there was miscommunication about Figure 5 f (comment 3).

What I was suggesting was to make the x axis of the plot exactly align with the position in the image. This would require more nuanced/thoughtful editing than just widening the plot, since the robot motion doesn't end exactly at the right side of the image. Either re-cropping the figure or changing the x-axis of the plot to start a little negative and end slightly after 200, so that the leftmost robot image is at $x=0$, and rightmost robot image is at $x=200$.

Response: Figure 5 f is adjusted according to the comment.